# A Simple Yet Effective Strategy to Robustify the Meta Learning Paradigm

**Qi Wang**[1*]    **Yiqin Lv**[1*]    **Yanghe Feng**[2]    **Zheng Xie**[1†]    **Jincai Huang**[2†]

[1]College of Science, National University of Defense Technology
[2]College of Systems Engineering, National University of Defense Technology
{wangqi15,lvyiqin98,fengyanghe,xiezheng81,huangjincai}@nudt.edu.cn

## Abstract

Meta learning is a promising paradigm to enable skill transfer across tasks. Most previous methods employ the empirical risk minimization principle in optimization. However, the resulting worst fast adaptation to a subset of tasks can be catastrophic in risk-sensitive scenarios. To robustify fast adaptation, this paper optimizes meta learning pipelines from a distributionally robust perspective and meta trains models with the measure of expected tail risk. We take the two-stage strategy as heuristics to solve the robust meta learning problem, controlling the worst fast adaptation cases at a certain probabilistic level. Experimental results show that our simple method can improve the robustness of meta learning to task distributions and reduce the conditional expectation of the worst fast adaptation risk.

## 1   Introduction

The past decade has witnessed the remarkable progress of deep learning in real-world applications (LeCun et al., 2015). However, training deep learning models requires an enormous dataset and intensive computational power. At the same time, these pre-trained models can frequently encounter deployment difficulties when the dataset's distribution drifts in testing time (Lesort et al., 2021).

As a result, the paradigm of *meta learning* or *learning to learn* is proposed and impacts the machine learning scheme (Finn et al., 2017), which leverages past experiences to enable fast adaptation to unseen tasks. Moreover, in the past few

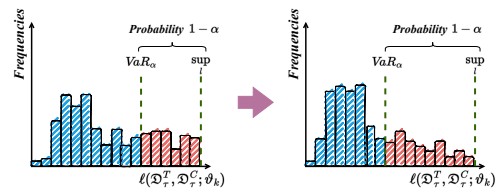

Figure 1: **Illustrations of Distributionally Robust Fast Adaptation.** Shown are histograms of meta risk function values $\ell(\mathfrak{D}_\tau^T, \mathfrak{D}_\tau^C; \vartheta)$ in the task distribution $p(\tau)$. Given a probability $\alpha$, we optimize meta learning model parameters $\vartheta$ to decrease the risk quantity $\text{CVaR}_\alpha$ in Definition (3).

years, there has grown a large body of meta learning methods to find plausible strategies to distill common knowledge into separate tasks (Finn et al., 2017; Duan et al., 2016; Garnelo et al., 2018a).

Notably, most previous work concentrates merely on the fast adaptation strategies and employs the standard risk minimization principle, *e.g.* the empirical risk minimization, ignoring the difference between tasks in fast adaptation. Given the sampled batch from the task distribution, the standard meta learning methods weight tasks equally in fast adaptation. Such an implementation raises concerns in some real-world scenarios, when worst fast adaptation is catastrophic in a range of risk-sensitive applications (Johannsmeier et al., 2019; Jaafra et al., 2019; Wang et al., 2023b). For example, in

---

[*]These authors contributed equally.

[†]Zheng Xie and Jincai Huang are corresponding authors.

37th Conference on Neural Information Processing Systems (NeurIPS 2023).

robotic manipulations, humanoid robots (Duan, 2017) can quickly leverage past motor primitives to walk on plain roads but might suffer from tribulation doing this on rough roads.

**Research Motivations.** Instead of seeking novel fast adaptation strategies, we take more interest in optimization principles for meta learning. Given the meta trained model, this paper stresses the performance difference in fast adaptation to various tasks as an indispensable consideration. As *the concept of robustness in fast adaptation* has not been sufficiently explored from the task distribution perspective, researching this topic has more practical significance and deserves more attention in meta learning. Naturally, we raise the question below:

*Can we reconsider the meta learning paradigm through the lens of risk distribution, and are there plausible measures to enhance the fast adaptation robustness in some vulnerable scenarios?*

**Developed Methods.** In an effort to address the above concerns and answer these questions, *we reduce robust fast adaptation in meta learning to a stochastic optimization problem within the principle of minimizing the expected tail risk, e.g., conditional value-at-risk (CVaR) (Rockafellar et al., 2000)*. To tractably solve the problem, we adopt a two-stage heuristic strategy for optimization with the help of crude Monte Carlo methods (Kroese and Rubinstein, 2012) and give some theoretical analysis. In each optimization step, the algorithm estimates the value-at-risk (VaR) (Rockafellar et al., 2000) from a meta batch of tasks and screens and optimizes a percentile of task samples vulnerable to fast adaptation. As illustrated in Fig. (1), such an operation is equivalent to iteratively reshaping the task risk distribution to increase robustness. The consequence of optimizing the risk function distributions $\vartheta_k \to \vartheta_{k+1}$ is to transport the probability mass in high-risk regions to the left side gradually. In this manner, the distribution of risk functions in the task domain can be optimized toward the anticipated direction that controls the worst-case fast adaptation at a certain probabilistic level.

**Outline & Primary Contributions.** We overview related meta learning and robust optimization work in Section (2). Section (3) introduces general notations and describes meta learning optimization objectives together with typical models. The distributionally robust meta learning problem is presented together with a heuristic optimization strategy in Section (4). We report experimental results and analysis in Section (5), followed by conclusions and limitations in Section (6). **Our primary contribution** is two-fold:

1. We recast the robustification of meta learning to a distributional optimization problem. The resulting framework minimizes the conditional expectation of task risks, namely the tail risk, which unifies vanilla meta-learning and worst-case meta learning frameworks.

2. To resolve the robust meta learning problem, we adopt the heuristic two-stage strategy and demonstrate its improvement guarantee. Experimental results show the effectiveness of our method, enhancing fast adaptation robustness and mitigating the worst-case performance.

## 2 Literature Review

**Meta Learning Methods.** In practice, meta learning enables fast learning (adaptation to unseen tasks) via slow learning (meta training in a collection of tasks). There exist different families of meta learning methods. The optimization-based methods, such as model agnostic meta learning (MAML) (Finn et al., 2017) and its variants (Finn et al., 2018; Rajeswaran et al., 2019; Grant et al., 2018; Vuorio et al., 2019; Abbas et al., 2022), try to find the optimal initial parameters of models and then execute gradient updates over them to achieve adaptation with a few examples. The context-based methods, *e.g.* conditional neural processes (CNPs) (Garnelo et al., 2018a), neural processes (NPs) (Garnelo et al., 2018b) and extensions (Gordon et al., 2019; Foong et al., 2020; Wang and Van Hoof, 2020; Gondal et al., 2021; Wang and van Hoof, 2022; Shen et al., 2021; Wang et al., 2023a), learn the representation of tasks in the function space and formulate meta learning models as exchangeable stochastic processes. The metrics-based methods (Snell et al., 2017; Allen et al., 2019; Bartunov and Vetrov, 2018) embed tasks in a metric space and can achieve competitive performance in few-shot classification tasks. Other methods like memory-augmented models (Santoro et al., 2016; Xiao et al., 2021, 2022), recurrent models (Duan et al., 2016) and hyper networks (Zhao et al., 2020; Beck et al., 2023) are also modified for meta learning purposes.

Besides, there exist several important works investigating the generalization capability of methods. Bai et al. (2021) conduct the theoretical analysis of the train-validation split and connect it to

optimality. In (Chen et al., 2021), a generalization bound is constructed for MAML through the lens of information theory. Denevi et al. (2019) study an average risk bound and estimate the bias for improving stochastic gradient optimization in meta learning.

**Robust Optimization.** When performing robust optimization for downstream tasks in deep learning, we can find massive work concerning the adversarial input noise (Goodfellow et al., 2018; Goel et al., 2020; Ren et al., 2021), or the perturbation on the model parameters (Goodfellow et al., 2014; Kurakin et al., 2016; Liu et al., 2018; Silva and Najafirad, 2020). In contrast, this paper studies the robustness of fast adaptation in meta learning. In terms of robust principles, the commonly considered one is the worst-case optimization (Olds, 2015; Zhang et al., 2020; Tay et al., 2022). For example, Collins et al. (2020) conducts the worst-case optimization in MAML to obtain the robust meta initialization. Considering the influence of adversarial examples, Goldblum et al. (2019) propose to adversarially meta train the model for few-shot image classification. Wang et al. (2020) adopt the worst-case optimization in MAML to increase the model robustness by injecting adversarial noise to the input. *However, distributionally robust optimization (Rahimian and Mehrotra, 2019) is rarely examined in the presence of the meta learning task distribution.*

## 3 Preliminaries

**Notations.** Consider the distribution of tasks $p(\tau)$ for meta learning and denote the task space by $\Omega_\tau$. Let $\tau$ be a task sampled from $p(\tau)$ with $\mathcal{T}$ the set of all tasks. We denote the meta dataset by $\mathfrak{D}_\tau$. For example, in few-shot regression problems, $\mathfrak{D}_\tau$ refers to a set of data points $\{(x_i, y_i)\}_{i=1}^m$ to fit.

Generally, $\mathfrak{D}_\tau$ are processed into the context set $\mathfrak{D}_\tau^C$ for fast adaptation, and the target set $\mathfrak{D}_\tau^T$ for evaluating adaptation performance. As an instance, we process the dataset $\mathfrak{D}_\tau = \mathfrak{D}_\tau^C \cup \mathfrak{D}_\tau^T$ with a fixed partition in MAML (Finn et al., 2017). $\mathfrak{D}_\tau^C$ and $\mathfrak{D}_\tau^T$ are respectively used for the inner loop and the outer loop in model optimization.

**Definition 1 (Meta Risk Function)** *With the task $\tau \in \mathcal{T}$ and the pre-processed dataset $\mathfrak{D}_\tau$ and the model parameter $\vartheta \in \Theta$, the meta risk function is a map $\ell : \mathfrak{D}_\tau \times \Theta \mapsto \mathbb{R}^+$.*

In meta learning, the meta risk function $\ell(\mathfrak{D}_\tau^T, \mathfrak{D}_\tau^C; \vartheta)$, *e.g.* instantiations in Example (1)/(2), is to evaluate the model performance after fast adaptation. Now we turn to the commonly-used risk minimization principle, which plays a crucial role in fast adaptation. To summarize, we include the vanilla and worst-case optimization objectives as follows.

**Expected Risk Minimization for Meta Learning.** The objective that applies to most vanilla meta learning methods can be formulated in Eq. (1), and the optimization executes in a distribution over tasks $p(\tau)$. The Monte Carlo estimate corresponds to the *empirical risk minimization* principle.

$$\min_{\vartheta \in \Theta} \mathcal{E}(\vartheta) := \mathbb{E}_{p(\tau)} \left[ \ell(\mathfrak{D}_\tau^T, \mathfrak{D}_\tau^C; \vartheta) \right] \tag{1}$$

Here $\vartheta$ is the parameter of meta learning models, which includes parameters for common knowledge shared across all tasks and for fast adaptation. Furthermore, the task distribution heavily influences the direction of optimization in meta training.

**Worst-case Risk Minimization for Meta Learning.** This also considers meta learning in the task distribution $p(\tau)$, but the worst case in fast adaptation is the top priority in optimization.

$$\min_{\vartheta \in \Theta} \max_{\tau \in \mathcal{T}} \ell(\mathfrak{D}_\tau^T, \mathfrak{D}_\tau^C; \vartheta) \tag{2}$$

The optimization objective is built upon the min-max framework, advancing the robustness of meta learning to the worst case. Approaches like TR-MAML (Collins et al., 2020) sample the worst task in a batch to meta train with gradient updates. Nevertheless, this setup might result in a highly conservative solution where the worst case only happens with an incredibly lower chance.

# 4 Distributionally Robust Fast Adaptation

This section starts with the concept of risk measures and the derived meta learning optimization objective. Then a heuristic strategy is designed to approximately solve the problem. Finally, we provide two examples of distributionally robust meta learning methods.

## 4.1 Meta Risk Functions as Random Variables

**Assumption 1** *The meta risk function $\ell(\mathfrak{D}_\tau^T, \mathfrak{D}_\tau^C; \vartheta)$ is $\beta_\tau$-Lipschitz continuous w.r.t. $\vartheta$, which suggests: there exists a positive constant $\beta_\tau$ such that $\forall \{\vartheta, \vartheta'\} \in \Theta$:*

$$|\ell(\mathfrak{D}_\tau^T, \mathfrak{D}_\tau^C; \vartheta) - \ell(\mathfrak{D}_\tau^T, \mathfrak{D}_\tau^C; \vartheta')| \leq \beta_\tau ||\vartheta - \vartheta'||.$$

Let $(\Omega_\tau, \mathcal{F}_\tau, \mathbb{P}_\tau)$ denote a probability measure over the task space, where $\mathcal{F}_\tau$ corresponds to a $\sigma$-algebra on the subsets of $\Omega_\tau$. And we have $(\mathbb{R}^+, \mathcal{B})$ a probability measure over the non-negative real domain for the previously mentioned meta risk function $\ell(\mathfrak{D}_\tau^T, \mathfrak{D}_\tau^C; \vartheta)$ with $\mathcal{B}$ a Borel $\sigma$-algebra. For any $\vartheta \in \Theta$, the meta learning operator $\mathcal{M}_\vartheta : \Omega_\tau \mapsto \mathbb{R}^+$ is defined as:

$$\mathcal{M}_\vartheta : \tau \mapsto \ell(\mathfrak{D}_\tau^T, \mathfrak{D}_\tau^C; \vartheta).$$

In this way, $\ell(\mathfrak{D}_\tau^T, \mathfrak{D}_\tau^C; \vartheta)$ can be viewed as a random variable to induce the distribution $p(\ell(\mathfrak{D}_\tau^T, \mathfrak{D}_\tau^C; \vartheta))$. Further, the cumulative distribution function can be formulated as $F_\ell(l; \vartheta) = \mathbb{P}(\{\ell(\mathfrak{D}_\tau^T, \mathfrak{D}_\tau^C; \vartheta) \leq l; \tau \in \Omega_\tau, l \in \mathbb{R}^+\})$ w.r.t. the task space. Note that $F_\ell(l; \vartheta)$ implicitly depends on the model parameter $\vartheta$, and we cannot access a closed-form in practice.

**Definition 2 (Value-at-Risk)** *Given the confidence level $\alpha \in [0,1]$, the task distribution $p(\tau)$ and the model parameter $\vartheta$, the VaR (Rockafellar et al., 2000) of the meta risk function is defined as:*

$$VaR_\alpha \left[ \ell(\mathcal{T}, \vartheta) \right] = \inf_{l \in \mathbb{R}^+} \{l | F_\ell(l; \vartheta) \geq \alpha, \tau \in \mathcal{T}\}.$$

**Definition 3 (Conditional Value-at-Risk)** *Given the confidence level $\alpha \in [0,1]$, the task distribution $p(\tau)$ and the model parameter $\vartheta$, we focus on the constrained domain of the random variable $\ell(\mathfrak{D}_\tau^T, \mathfrak{D}_\tau^C; \vartheta)$ with $\ell(\mathfrak{D}_\tau^T, \mathfrak{D}_\tau^C; \vartheta) \geq VaR_\alpha[\ell(\mathcal{T}, \vartheta)]$. The conditional expectation of this is termed as conditional value-at-risk (Rockafellar et al., 2000):*

$$CVaR_\alpha \left[ \ell(\mathcal{T}, \vartheta) \right] = \int_0^\infty l dF_\ell^\alpha(l; \vartheta),$$

*where the normalized cumulative distribution is as follows:*

$$F_\ell^\alpha(l; \vartheta) = \begin{cases} 0, & l < VaR_\alpha[\ell(\mathcal{T}, \vartheta)] \\ \frac{F_\ell(l; \vartheta) - \alpha}{1 - \alpha}, & l \geq VaR_\alpha[\ell(\mathcal{T}, \vartheta)]. \end{cases}$$

This results in the normalized probability measure $(\Omega_{\alpha,\tau}, \mathcal{F}_{\alpha,\tau}, \mathbb{P}_{\alpha,\tau})$ over the task space, where $\Omega_{\alpha,\tau} := \bigcup_{\ell \geq VaR_\alpha[\ell(\mathcal{T}, \vartheta)]} \left[ \mathcal{M}_\vartheta^{-1}(\ell) \right]$. For ease of presentation, we denote the corresponding task distribution constrained in $\Omega_{\alpha,\tau}$ by $p_\alpha(\tau; \vartheta)$.

Rather than optimizing $VaR_\alpha$, a quantile, in meta learning, we take more interest in $CVaR_\alpha$ optimization, a type of *the expected tail risk*. Such risk measure regards the conditional expectation and has more desirable properties for meta learning: *more adequate in handling adaptation risks in extreme tails*, *more accessible sensitivity analysis w.r.t. $\alpha$*, and *more efficient optimization*.

**Remark 1** *$CVaR_\alpha \left[ \ell(\mathcal{T}, \vartheta) \right]$ to minimize is respectively equivalent with the vanilla meta learning optimization objective in Eq. (1) when $\alpha = 0$ and the worst-case meta learning optimization objective in Eq. (3) when $\alpha$ is sufficiently close to 1.*

## 4.2 Meta Learning via Controlling the Expected Tail Risk

As mentioned in Remark (1), the previous two meta learning objectives can be viewed as special cases within the $CVaR_\alpha$ principle. Furthermore, we turn to a particular distributionally robust fast

adaptation with the adjustable confidence level $\alpha$ to control the expected tail risk in optimization as follows.

**Distributionally Robust Meta Learning Objective.** With the previously introduced normalized probability density function $p_\alpha(\tau; \vartheta)$, minimizing $\text{CVaR}_\alpha[\ell(\mathcal{T}, \vartheta)]$ can be rewritten as Eq. (3).

$$\min_{\vartheta \in \Theta} \mathcal{E}_\alpha(\vartheta) := \mathbb{E}_{p_\alpha(\tau; \vartheta)} \left[ \ell(\mathfrak{D}_\tau^T, \mathfrak{D}_\tau^C; \vartheta) \right] \tag{3}$$

Even though $\text{CVaR}_\alpha[\ell(\mathcal{T}, \vartheta)]$ is a function of the model parameter $\vartheta$, the integral in Eq. (3) is intractable due to the involvement of $p_\alpha(\tau; \vartheta)$ in a non-closed form.

**Assumption 2** *For meta risk function values, the cumulative distribution function $F_\ell(l; \vartheta)$ is $\beta_\ell$-Lipschitz continuous w.r.t. $l$, and the implicit normalized probability density function of tasks $p_\alpha(\tau; \vartheta)$ is $\beta_\theta$-Lipschitz continuous w.r.t. $\vartheta$.*

**Assumption 3** *For any valid $\vartheta \in \Theta$ and corresponding implicit normalized probability density function of tasks $p_\alpha(\tau; \vartheta)$, the meta risk function value can be bounded by a positive constant $\mathcal{L}_{\max}$:*

$$\sup_{\tau \in \Omega_{\alpha, \tau}} \ell(\mathfrak{D}_{\tau_i}^T, \mathfrak{D}_{\tau_i}^C; \vartheta) \leq \mathcal{L}_{\max}.$$

**Proposition 1** *Under assumptions (1)/(2)/(3), the meta learning optimization objective $\mathcal{E}_\alpha(\vartheta)$ in Eq. (3) is continuous w.r.t. $\vartheta$.*

Further, we use $\xi_\alpha(\vartheta)$ to denote the $\text{VaR}_\alpha[\ell(\mathcal{T}, \vartheta)]$ for simple notations. The same as that in (Rockafellar et al., 2000), we introduce a slack variable $\xi \in \mathbb{R}$ and the auxiliary risk function $\left[ \ell(\mathfrak{D}_\tau^T, \mathfrak{D}_\tau^C; \vartheta) - \xi \right]^+ := \max\{\ell(\mathfrak{D}_\tau^T, \mathfrak{D}_\tau^C; \vartheta) - \xi, 0\}$. To circumvent directly optimizing the non-analytical $p_\alpha(\tau; \vartheta)$, we can convert the probability constrained function $\mathcal{E}_\alpha(\vartheta)$ to the below unconstrained one after optimizing $\xi$:

$$\varphi_\alpha(\xi; \vartheta) = \xi + \frac{1}{1 - \alpha} \mathbb{E}_{p(\tau)} \left[ \left[ \ell(\mathfrak{D}_\tau^T, \mathfrak{D}_\tau^C; \vartheta) - \xi \right]^+ \right].$$

It is demonstrated that $\mathcal{E}_\alpha(\vartheta) = \min_{\xi \in \mathbb{R}} \varphi_\alpha(\xi; \vartheta)$ and $\xi_\alpha \in \arg\min_{\xi \in \mathbb{R}} \varphi_\alpha(\xi; \vartheta)$ in (Rockafellar et al., 2000), and also note that $\text{CVaR}_\alpha$ is the upper bound of $\xi_\alpha$, implying

$$\xi_\alpha \leq \varphi_\alpha(\xi_\alpha; \vartheta) \leq \varphi_\alpha(\xi; \vartheta), \quad \forall \xi \in \mathbb{R} \text{ and } \forall \vartheta \in \Theta. \tag{4}$$

With the deductions from Eq.s (3)/(4), we can resort the distributionally robust meta learning optimization objective with the probability constraint into a unconstrained optimization objective as Eq.(5).

$$\min_{\vartheta \in \Theta, \xi \in \mathbb{R}} \xi + \frac{1}{1 - \alpha} \mathbb{E}_{p(\tau)} \left[ \left[ \ell(\mathfrak{D}_\tau^T, \mathfrak{D}_\tau^C; \vartheta) - \xi \right]^+ \right] \tag{5}$$

**Sample Average Approximation.** For the stochastic programming problem above, it is mostly intractable to derive the analytical form of the integral. Hence, we need to perform Monte Carlo estimates of Eq. (5) to obtain Eq. (6) for optimization.

$$\min_{\vartheta \in \Theta, \xi \in \mathbb{R}} \xi + \frac{1}{(1 - \alpha)\mathcal{B}} \sum_{i=1}^{\mathcal{B}} \left[ \ell(\mathfrak{D}_{\tau_i}^T, \mathfrak{D}_{\tau_i}^C; \vartheta) - \xi \right]^+ \tag{6}$$

**Remark 2** *If $\ell(\mathfrak{D}_\tau^T, \mathfrak{D}_\tau^C; \vartheta)$ is convex w.r.t. $\vartheta$, then Eq.s (5)/(6) are also convex functions. In this case, the optimization objective Eq. (6) of our interest can be resolved with the help of several convex programming algorithms (Fan et al., 2017; Meng et al., 2020; Levy et al., 2020).*

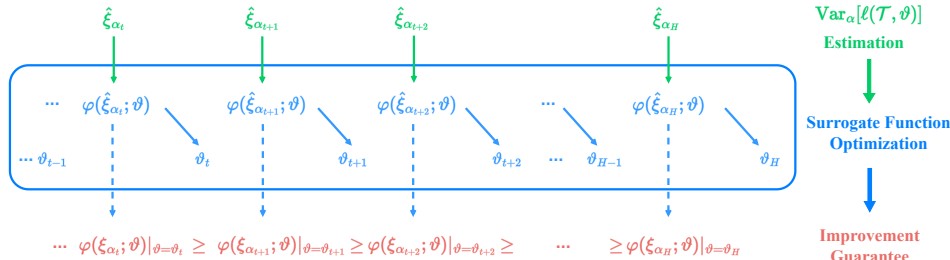

Figure 2: **Optimization Diagram of Distributionally Robust Meta Learning with Surrogate Functions.** From left to right: the meta model parameters $\vartheta$ in the middle block are optimized *w.r.t.* the constructed surrogate function $\varphi(\hat{\xi}_{\alpha_t}; \vartheta)$ marked in blue in $t$-th iteration. Under certain conditions in Theorem (1), the distributionally robust meta learning objective $\varphi(\xi_\alpha; \vartheta)$ marked in pink can be decreased monotonically until it reaches the convergence in the $H$-th iteration.

## 4.3 Heuristic Algorithms for Optimization

Unfortunately, most existing meta learning models' risk functions (Finn et al., 2017; Garnelo et al., 2018a; Santoro et al., 2016; Li et al., 2017; Duan et al., 2016), $\ell(\mathfrak{D}_\tau^T, \mathfrak{D}_\tau^C; \vartheta)$ are non-convex *w.r.t.* $\vartheta$, bringing difficulties in optimization of Eq.s (5)/(6).

To this end, we propose a simple yet effective optimization strategy, where the sampled task batch is used to approximate the $\mathrm{VaR}_\alpha$ and the integral in Eq. (5) for deriving Eq. (6). In detail, two stages are involved in iterations: *(i) approximate $VaR_\alpha[\ell(\mathcal{T}, \vartheta)] \approx \hat{\xi}_\alpha$ with the meta batch values, which can be achieved via either a quantile estimator (Dong and Nakayama, 2018) or other density estimators; (ii) optimize $\vartheta$ in Eq. (6) via stochastic updates after replacing $\xi_\alpha$ by the estimated $\hat{\xi}_\alpha$.*

**Proposition 2** *Suppose there exists $\delta \in \mathbb{R}^+$ such that $|\xi_\alpha(\vartheta) - \hat{\xi}_\alpha(\vartheta)| < \delta$ with $\hat{\xi}_\alpha(\vartheta)$ an estimate of $\xi_\alpha(\vartheta)$. Then there exists a constant $\kappa_\alpha = \max\{\frac{2-\alpha}{1-\alpha}, \frac{\alpha}{1-\alpha}\}$ such that*

$$\varphi_\alpha(\hat{\xi}_\alpha(\vartheta); \vartheta) - \kappa_\alpha\delta < \mathcal{E}_\alpha(\vartheta) \leq \varphi_\alpha(\hat{\xi}_\alpha(\vartheta); \vartheta).$$

The performance gap resulting from $\mathrm{VaR}_\alpha$ approximation error is estimated in Proposition (2). For ease of implementation, we adopt crude Monte Carlo methods (Kroese and Rubinstein, 2012) to obtain a consistent estimator of $\xi_\alpha$.

**Theorem 1 (Improvement Guarantee)** *Under assumptions (1)/(2)/(3), suppose that the estimate error with the crude Monte Carlo holds: $|\hat{\xi}_{\alpha_t} - \xi_{\alpha_t}| \leq \frac{\lambda}{\beta_\ell(1-\alpha)^2}, \forall t \in \mathbb{N}^+$, with the subscript $t$ the iteration number, $\lambda$ the learning rate in stochastic gradient descent, $\beta_\ell$ the Lipschitz constant of the risk cumulative distribution function, and $\alpha$ the confidence level. Then the proposed heuristic algorithm with the crude Monte Carlo can produce at least a local optimum for distributionally robust fast adaptation.*

Note that Theorem (1) indicates that under certain conditions, using the above heuristic algorithm has the performance improvement guarantee, which corresponds to Fig. (2). The error resulting from the approximate algorithm is further estimated in Appendix Theorem (2).

## 4.4 Instantiations & Implementations

Our proposed optimization strategy applies to all meta learning methods and has the improvement guarantee in Theorem (1). Here we take two representative meta learning methods, MAML (Finn et al., 2017) and CNP (Garnelo et al., 2018a), as examples and show how to robustify them through the lens of risk distributions. Note that the forms of $\ell(\mathfrak{D}_\tau^T, \mathfrak{D}_\tau^C; \vartheta)$ sometimes differ in these methods. Also, the detailed implementation of the strategy relies on specific meta learning algorithms and models.

**Example 1 (DR-MAML)** *With the task distribution $p(\tau)$ and model agnostic meta learning (Finn et al., 2017), the distributionally robust MAML treats the meta learning problem as a bi-level*

*optimization with a VaR$_\alpha$ relevant constraint.*

$$\min_{\substack{\vartheta \in \Theta \\ \xi \in \mathbb{R}}} \xi + \frac{1}{1-\alpha} \mathbb{E}_{p(\tau)} \left[ \left[ \ell(\mathfrak{D}_\tau^T; \vartheta - \lambda \nabla_\vartheta \ell(\mathfrak{D}_\tau^C; \vartheta)) - \xi \right]^+ \right] \tag{7}$$

*The gradient operation $\nabla_\vartheta \ell(\mathfrak{D}_\tau^C; \vartheta)$ corresponds to the inner loop with the learning rate $\lambda$.*

The resulting distributionally robust MAML (DR-MAML) is still a optimization-based method, where a fixed percentage of tasks are screened for the outer loop. As shown in Fig. (3) and Eq. (7), the objective is to obtain a robust meta initialization of the model parameter $\vartheta$.

**Example 2 (DR-CNP)** *With the task distribution $p(\tau)$ and the conditional neural process (Garnelo et al., 2018a), the distributionally robust conditional neural process learns the functional representations with a CVaR$_\alpha$ constraint.*

$$\min_{\substack{\vartheta \in \Theta \\ \xi \in \mathbb{R}}} \xi + \frac{1}{1-\alpha} \mathbb{E}_{p(\tau)} \left[ \left[ \ell(\mathfrak{D}_\tau^T; z, \theta_2)) - \xi \right]^+ \right] \tag{8}$$

$$s.t. \ z = h_{\theta_1}(\mathfrak{D}_\tau^C) \text{ with } \vartheta = \{\theta_1, \theta_2\}$$

*Here $h_{\theta_1}$ is a set encoder network with $\theta_2$ the parameter of the decoder network.*

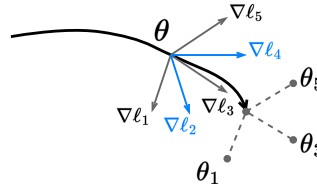

Figure 3: **Diagram of Distributionally Robust Fast Adaptation for Model Agnostic Meta Learning (Finn et al., 2017).** For example, with the size of the meta batch 5 and $\alpha = 40\%$, $5 * (1 - \alpha)$ tasks in gray with the worst fast adaptation performance are screened for updating meta initialization.

The resulting distributionally robust CNP (DR-CNP) is to find a robust functional embedding to induce underlying stochastic processes. Still, in Eq. (8), a proportion of tasks with the worst functional representation performance are used in meta training.

Moreover, we convey the pipelines of optimizing these developed distributionally robust models in Appendix **Algorithms (1)/(2)**.

## 5 Experimental Results and Analysis

This section presents experimental results and examines fast adaptation performance in a distributional sense. Without loss of generality, we take DR-MAML in Example (1) to run experiments.

**Benchmarks.** The same as work in (Collins et al., 2020), we use two commonly-seen downstream tasks for meta learning experiments: few-shot regression and image classification. Besides, ablation studies are included to assess other factors' influence or the proposed strategy's scalability.

**Baselines & Evaluations.** Since the primary investigation is regarding risk minimization principles, we consider the previously mentioned *expected risk minimization*, *worst-case minimization*, and *expected tail risk minimization* for meta learning. Hence, MAML (empirical risk), TR-MAML (worst-case risk), and DR-MAML (expected tail risk) serve as examined methods. We evaluate these methods' performance based on the *Average*, *Worst-case*, and *CVaR$_\alpha$* metrics. For the confidence level to meta train DR-MAML, we empirically set $\alpha = 0.7$ for few-shot regression tasks and $\alpha = 0.5$ image classification tasks without external configurations.

### 5.1 Sinusoid Regression

Following (Finn et al., 2017; Collins et al., 2020), we conduct experiments in sinusoid regression tasks. The mission is to approximate the function $f(x) = a\sin(x - b)$ with K-shot randomly sampled function points, where the task is defined by $a$ and $b$. In the sine function family, the target range, amplitude range, and phase range are

Table 1: **Test average mean square errors (MSEs) with reported standard deviations for sinusoid regression (5 runs).** We respectively consider 5-shot and 10-shot cases with $\alpha = 0.7$. The results are evaluated across the 490 meta-test tasks, as in (Collins et al., 2020). The best results are in bold.

| Method | 5-shot | | | 10-shot | | |
|---|---|---|---|---|---|---|
| | Average | Worst | CVaR$_\alpha$ | Average | Worst | CVaR$_\alpha$ |
| MAML (Finn et al., 2017) | 1.02$_{\pm 0.10}$ | 3.89$_{\pm 0.83}$ | 2.25$_{\pm 0.15}$ | 0.66$_{\pm 0.16}$ | 2.57$_{\pm 0.70}$ | 1.15$_{\pm 0.19}$ |
| TR-MAML (Collins et al., 2020) | 1.09$_{\pm 0.08}$ | **2.28$_{\pm 0.35}$** | 1.79$_{\pm 0.06}$ | 0.77$_{\pm 0.11}$ | **1.68$_{\pm 0.43}$** | 1.27$_{\pm 0.28}$ |
| DR-MAML (Ours) | **0.89$_{\pm 0.04}$** | 2.91$_{\pm 0.46}$ | **1.76$_{\pm 0.02}$** | **0.54$_{\pm 0.01}$** | 1.70$_{\pm 0.17}$ | **0.96$_{\pm 0.01}$** |

respectively $[-5.0, 5.0] \subset \mathbb{R}$, $a \in [0.1, 5.0]$ and $b \in [0, 2\pi]$. In the setup of meta training and testing datasets, a distributional shift exists: numerous easy tasks and several difficult tasks are generated to formulate the training dataset with all tasks in the space as the testing one. Please refer to Appendix (J) for a detailed partition of meta-training, testing tasks, and neural architectures.

**Result Analysis.** We list meta-testing MSEs in sinusoid regression in Table (1). As expected, the tail risk minimization principle in DR-MAML can lead to an intermediate performance in the worst-case. In both cases, the comparison between MAML and DR-MAML in MSEs indicates that such probabilistic-constrained optimization in the task space even has the potential to advance average fast adaptation performance. In contrast, TR-MAML has to sacrifice more average performance for the worst-case improvement.

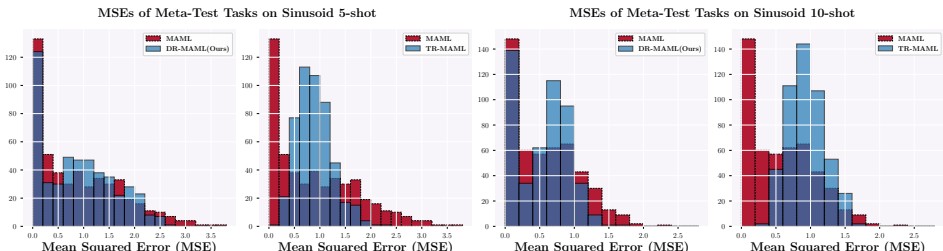

Figure 4: **Histograms of Meta-Testing Performance in Sinusoid Regression Problems.** With $\alpha = 0.7$, we respectively visualize the comparison results, DR-MAML-vs-MAML and TR-MAML-vs-MAML in 5-shot (Two Sub-figures Left Side) and 10-shot (Two Sub-figures Right Side) cases, for a sample trial.

More intuitively, Fig. (4) illustrates MSE statistics on the testing task distribution and further verifies the effect of the $\text{CVaR}_\alpha$ principle in decreasing the proportional worst-case errors. In 5-shot cases, the difference in MSE statistics is more significant: *the worst-case method tends to increase the skewness in risk distributions* with many task risk values gathered in regions of intermediate performance, which is unfavored in general cases. As for why DR-MAML surpasses MAML in terms of average performance, we attribute it to the benefits of external robustness in several scenarios, *e.g.*, the drift of training/testing task distributions.

Table 2: **Average N-way K-shot classification accuracies in Omniglot with reported standard deviations (3 runs).** With $\alpha = 0.5$, the best results are in bold.

| | Meta-Training Alphabets | | | | | | Meta-Testing Alphabets | | | | | |
| | 5-way 1-shot | | | 20-way 1-shot | | | 5-way 1-shot | | | 20-way 1-shot | | |
| Method | Average | Worst | $\text{CVaR}_\alpha$ | Average | Worst | $\text{CVaR}_\alpha$ | Average | Worst | $\text{CVaR}_\alpha$ | Average | Worst | $\text{CVaR}_\alpha$ |
|---|---|---|---|---|---|---|---|---|---|---|---|---|
| MAML (Finn et al., 2017) | **98.4**±0.2 | 82.4±1.1 | **96.9**±0.5 | 99.2±0.1 | 33.9±3.0 | 80.9±0.7 | 93.5±0.2 | 82.5±0.2 | 91.6±0.6 | 67.6±2.0 | 49.7±3.5 | 60.4±1.7 |
| TR-MAML (Collins et al., 2020) | 97.4±0.6 | **95.0**±0.3 | 96.5±0.4 | 92.2±0.8 | **82.4**±2.1 | **87.2**±0.9 | 93.1±1.1 | **85.3**±1.9 | 91.3±0.9 | 74.3±1.4 | **58.4**±1.8 | **68.5**±1.2 |
| DR-MAML (Ours) | 97.1±0.3 | 84.0±0.4 | 95.1±0.3 | **99.6**±0.6 | 57.9±2.4 | 84.8±0.7 | **93.7**±0.4 | 84.1±0.8 | **92.1**±0.5 | **74.6**±1.2 | 51.0±2.3 | 66.4±1.4 |

## 5.2 Few-Shot Image Classification

Here we do investigations in few-shot image classification. Each task is an N-way K-shot classification with N the number of classes and K the number of labeled examples in one class. The Omniglot (Lake et al., 2015) and *mini*-ImageNet (Vinyals et al., 2016) datasets work as benchmarks for examination. We retain the setup of datasets in work (Collins et al., 2020).

Table 3: **Average 5-way 1-shot classification accuracies in *mini*-ImageNet with reported standard deviations (3 runs).** With $\alpha = 0.5$, the best results are in bold.

| | Eight Meta-Training Tasks | | | Four Meta-Testing Tasks | | |
| Method | Average | Worst | $\text{CVaR}_\alpha$ | Average | Worst | $\text{CVaR}_\alpha$ |
|---|---|---|---|---|---|---|
| MAML (Finn et al., 2017) | 70.1±2.2 | 48.0±4.5 | 63.2±2.6 | 46.6±0.4 | 44.7±0.7 | 44.6±0.7 |
| TR-MAML (Collins et al., 2020) | 63.2±1.3 | 60.7±1.6 | 62.1±1.2 | 48.5±0.6 | 45.9±0.8 | 46.6±0.5 |
| DR-MAML (Ours) | **70.2**±0.2 | **63.4**±0.2 | **67.2**±0.1 | **49.4**±0.1 | **47.1**±0.1 | **47.5**±0.1 |

**Result Analysis.** The classification accuracies in Omniglot are illustrated in Table (2): In 5-way 1-shot cases, DR-MAML obtains the best average and $\text{CVaR}_\alpha$ in meta-testing datasets, while TR-MAML achieves the best worst-case performance with a slight degradation of average performance compared to MAML in both training/testing datasets. In 20-way 1-shot cases, for training/testing datasets, we surprisingly notice that the expected tail risk is not well optimized with DR-MAML, but there is an average performance gain; while TR-MAML works best in worst-case/$\text{CVaR}_\alpha$ metrics.

When it comes to *mini*-ImageNet, findings are distinguished a lot from the above one: in Table (3), DR-MAML yields the best result in all evaluation metrics and cases, even regarding the worst-case. TR-MAML can also improve all metrics in meta-testing cases. Overall, challenging meta-learning tasks can reveal more advantages of DR-MAML over others.

## 5.3 Ablation Studies

This part mainly checks the influence of the confidence level $\alpha$, meta batch size, and other optimization strategies towards the distribution of fast adapted risk values. Apart from examining these factors of interest, additional experiments are also conducted in this paper; please refer to Appendix (K).

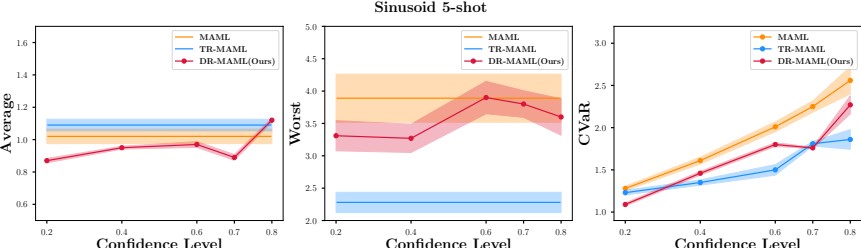

Figure 5: **Meta Testing MSEs of Meta-Trained DR-MAML with Various Confidence Levels $\alpha$.** MAML and TR-MAML are irrelevant with the variation of $\alpha$ in meta-training. The plots report testing MSEs with standard error bars in shadow regions.

**Sensitivity to Confidence Levels $\alpha$.** To deepen understanding of the confidence level $\alpha$'s effect in fast adaptation performance, we vary $\alpha$ to train DR-MAML and evaluate models under previously mentioned metrics. Taking the sinusoid `5-shot` regression as an example, we calculate MSEs and visualize the fluctuations with additional $\alpha$-values in Fig. (5). The results in the worst-case exhibit higher deviations. The trend in Average/CVaR$_\alpha$ metrics shows that with increasing $\alpha$, DR-MAML gradually approaches TR-MAML in average and CVaR$_\alpha$ MSEs. With $\alpha \leq 0.8$, ours is less sensitive to the confidence level and mostly beats MAML/TR-MAML in average performance. Though ours aims at optimizing CVaR$_\alpha$, it cannot always ensure such a metric to surpass TR-MAML in all confidence levels due to rigorous assumptions in Theorem (1).

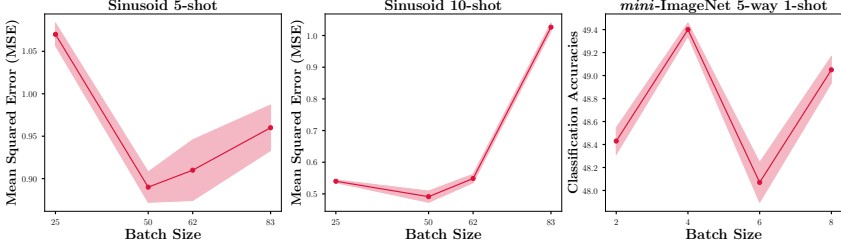

Figure 6: **Meta Testing Average Performance of Meta-Trained DR-MAML with Various Sizes of the Meta Batch.** The plots report average results with standard error bars in shadow regions.

**Influence of the Task Batch Size.** Note that our optimization strategy relies on the estimate of VaR$_\alpha$, and the improvement guarantee relates to the estimation error. Theoretically, the meta batch in training can directly influence the optimization result. Here we vary the meta batch size in training, and evaluated results are visualized in Fig. (6). In regression scenarios, the average MSEs can decrease to a certain level with an increase of meta batch, and then performance degrades. We attribute the performance gain with a batch increase to more accurate VaR$_\alpha$ estimates; however, a meta batch larger than some threshold can worsen the first-order meta learning algorithms' efficiency, similarly observed in (Nichol et al., 2018). As for classification scenarios, there appears no clear trend since the meta batch is smaller enough.

**Comparison with Other Optimization Strategies.** Note that instantiations of distributionally robust meta learning methods, such as DR-MAML and DR-CNPs in Examples (1)/(2) are regardless of optimization strategies and can be optimized via any heuristic algorithms for CVaR$_\alpha$ objectives.

Additionally, we use DR-MAML as the example and perform the comparison between our two-stage algorithm and the risk reweighted algorithm (Sagawa et al., 2020). The intuition of the risk reweighted algorithm is to relax the weights of tasks and assign more weights to the gradient of worst cases. The normalization of risk weights is achieved via the softmax operator. Though there is an improvement guarantee *w.r.t.* the probabilistic worst group of tasks, the algorithm is not specially designed for meta learning or CVaR$_\alpha$ objective.

$$\min_{\vartheta \in \Theta} \mathcal{E}_\alpha(\vartheta) := \mathbb{E}_{p(\tau;\vartheta)} \left[ \frac{p_\alpha(\tau;\vartheta)}{p(\tau;\vartheta)} \ell(\mathfrak{D}_\tau^T, \mathfrak{D}_\tau^C; \vartheta) \right] \approx \frac{1}{\mathcal{B}} \sum_{b=1}^{\mathcal{B}} \omega_b(\tau_b;\vartheta) \ell(\mathfrak{D}_{\tau_b}^T, \mathfrak{D}_{\tau_b}^C; \vartheta) \tag{9}$$

Table 4: **Test average mean square errors (MSEs) with reported standard deviations for sinusoid regression (5 runs). We mainly compare DR-MAML with different optimization algorithms.** `5-shot` and `10-shot` cases are respectively considered here. The results are evaluated across the 490 meta-test tasks, which is the same as in (Collins et al., 2020). With $\alpha = 0.7$ for meta training, the best testing results are in bold.

| Method | 5-shot | | | 10-shot | | |
|---|---|---|---|---|---|---|
| | Average | Worst | CVaR$_\alpha$ | Average | Worst | CVaR$_\alpha$ |
| DR-MAML (Group DRO, (Sagawa et al., 2020)) | 0.91$_{\pm 0.06}$ | 3.57$_{\pm 0.56}$ | 1.83$_{\pm 0.03}$ | 0.61$_{\pm 0.02}$ | 1.90$_{\pm 0.11}$ | 1.13$_{\pm 0.02}$ |
| DR-MAML (Two-Stage, Ours) | **0.89$_{\pm 0.04}$** | **2.91$_{\pm 0.46}$** | **1.76$_{\pm 0.02}$** | **0.54$_{\pm 0.01}$** | **1.70$_{\pm 0.17}$** | **0.96$_{\pm 0.01}$** |

Table 5: **Average** `5-way` `1-shot` **classification accuracies in** *mini*-**ImageNet with reported standard deviations (3 runs). We mainly compare DR-MAML with different optimization algorithms.** With $\alpha = 0.5$ for meta training, the best testing results are in bold.

| Method | Eight Meta-Training Tasks | | | Four Meta-Testing Tasks | | |
|---|---|---|---|---|---|---|
| | Average | Worst | CVaR$_\alpha$ | Average | Worst | CVaR$_\alpha$ |
| DR-MAML (Group DRO, (Sagawa et al., 2020)) | 67.0$_{\pm 0.2}$ | 56.6$_{\pm 0.4}$ | 61.6$_{\pm 0.2}$ | 49.1$_{\pm 0.2}$ | 46.6$_{\pm 0.1}$ | 47.2$_{\pm 0.2}$ |
| DR-MAML (Two-Stage, Ours) | **70.2$_{\pm 0.2}$** | **63.4$_{\pm 0.2}$** | **67.2$_{\pm 0.1}$** | **49.4$_{\pm 0.1}$** | **47.1$_{\pm 0.1}$** | **47.5$_{\pm 0.1}$** |

Meanwhile, the weight of task gradients after normalization is a biased estimator *w.r.t.* the constrained probability $p_\alpha(\tau; \vartheta)$ in the task space. In other words, the risk reweighted method can be viewed as approximation *w.r.t.* the importance weighted method in Eq. (9). In the importance weighted method, for tasks out of the region of $(1 - \alpha)$-proportional worst, the probability of sampling such tasks $\tau_b$ is zero, indicating $\omega_b(\tau_b; \vartheta) = 0$. While in risk reweighted methods, the approximate weight is assumed to satisfy $\omega_b(\tau_b; \vartheta) \propto \exp\left(\frac{\ell(\mathfrak{D}_{\tau_b}^T, \mathfrak{D}_{\tau_b}^C; \hat{\vartheta})}{\tau}\right)$, where $\hat{\vartheta}$ means last time updated meta model parameters and the risk function value is evaluated after fast adaptation.

In implementations, we keep the setup the same as Group DRO methods in (Sagawa et al., 2020) for meta-training. As illustrated in Table (4)/(5), DR-MAML with the two-stage optimization strategies consistently outperform that with the Group DRO ones in both `5-shot` and `10-shot` sinusoid cases regarding all metrics. The performance advantage of using the two-stage ones is not significant in *mini*-ImageNet scenarios. We can hypothesize that the estimate of VaR$_\alpha$ in continuous task domains, *e.g.*, sinusoid regression, is more accurate, and this probabilistically ensures the improvement guarantee with two-stage strategies. Both the VaR$_\alpha$ estimate in two-stage strategies and the importance weight estimate in the Group DRO ones may have a lot of biases in few-shot image classification, which lead to comparable performance.

## 6 Conclusion and Limitations

**Technical Discussions.** This work contributes more insights into robustifying fast adaptation in meta learning. Our utilized expected tail risk trades off the expected risk minimization and worst-case risk minimization, and the two-stage strategy works as the heuristic to approximately solve the problem with an improvement guarantee. Our strategy can empirically alleviate the worst-case fast adaptation and sometimes even improve average performance.

**Existing Limitations.** Though our robustification strategy is simple yet effective in implementations, empirical selection of the optimal meta batch size is challenging, especially for first-order optimization methods. Meanwhile, the theoretical analysis only applies to a fraction of meta learning tasks when risk function values are in a compact continuous domain.

**Future Extensions.** Designing a heuristic algorithm with an improvement guarantee is non-trivial and relies on the properties of risk functions. This research direction has practical meaning in the era of large models and deserves more investigations in terms of optimization methods. Also, establishing connections between the optimal meta batch size and specific stochastic optimization algorithms can be a promising theoretical research issue in this domain.

## Acknowledgments and Disclosure of Funding

This work is funded by National Natural Science Foundation of China (NSFC) with the Number # 62306326.

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

# Contents

# A  Frequently Asked Question

Here we collected technical questions and suggestions from researchers who helped check out the manuscript. We thank these researchers for precious questions and provide more details.

**Novelty & primary findings of this work.** Here we mainly summarize two points of novelty in this work:

- *Meta Learning Robustification Framework.* Though the concept of the expected tail risk has emerged for several decades and has been widely employed in financial domains, the application to fast adaptation or robustification of meta learning remains limited in literature as far as we know.

- *Optimization Strategy & Theoretical Analysis.* We theoretically analyze two-stage optimization strategies as the heuristic algorithm in optimizing the distributionally robust meta learning models and demonstrate the improvement guarantee under certain conditions.

As verified in experimental results in the main paper, placing a probabilistic constraint in the task space is meaningful. It circumvents the effect of over-pessimistic consideration (worst-case optimization), increases robustness in proportional cases, and mostly retains or even improves average performance.

Apart from the novelty in the framework and algorithm parts, we have several findings which bring crucial insights into meta learning: (i) Not all tasks are necessary to perform fast adaptation. (ii) Additional focus on the tail risk has the potential to enhance models' generalization capability. (iii) The tail risk instead of extreme worst-case risk can better advance robustness in challenging datasets.

**Meta risk function values as random variables.** In some few-shot learning related work, the context and the target dataset are equivalently called the support and query datasets. The definition of a task in meta learning is up to application scenarios and specific meta learning algorithms or models. The commonly-used sinusoid regression using model-agnostic meta learning (Finn et al., 2017) considers the fixed number of context points to induce tasks. In contrast, conditional neural processes (Garnelo et al., 2018a) for few-shot regression vary the number of context points to induce tasks. Once the context and the target are partitioned and the model parameter is specified, we can obtain a meta risk function value. However, the meta risk function values are not in a compact Euclidean subspace in several cases.

**The continuity of the meta risk probability density function.** This relates to the task distribution and meta learning problems. Throughout the few-shot regression task, the meta risk function value can be approximately viewed as a continuous random variable. However, when it comes to the few-shot image classification mission, the meta risk function value is impossible to cover the entire continuous interval. In these scenarios, the probable values of accuracies are finite. This makes the theoretical analysis, *e.g.* Theorem (1), no longer holds. For example, Assumption (2) will be unrealistic when there exists a constant gap between two accuracy values. Hence, we leave this part a future research direction in theoretical analysis. Regarding the meta risk function, ours shares the same setup as that in (Fallah et al., 2021). Admittedly, precisely modeling risk distributions can be crucial, and more techniques, *e.g.*, deep kernel estimators (Puchert et al., 2021) can be adopted to improve this point.

**The selection of baselines in robust meta learning.** A large body of prior work considers the robustness of meta learning in scenarios when the input of a data point, the model parameter, and the number of context points are trembled or modified. Robustness in presence of tasks distributions is seldom investigated except for the worst-case optimization in (Collins et al., 2020). Hence, we retain most of the setups in (Collins et al., 2020) for a fair comparison.

**Influence of risk minimization principles.** This paper is primarily devoted to studying the influence of the risk minimization principle on meta learning. The empirical risk minimization principle corresponds to reducing the Monte Carlo estimate of Average-case Meta Learning in this work. In comparison to TR-MAML (Collins et al., 2020), our approach has a couple of advantages as follows: (i) easier implementations. Note that min-max optimization is numerically unstable and requires a relaxation method for computationally intensive convex optimization, *e.g.*, robust stochastic mirror-prox algorithm used in TR-MAML (Collins et al., 2020). (ii) more flexible in terms of robustness concept. Theoretically, the worst-case meta learning corresponds to the extreme case of the distributionally robust risk minimization principle. (iii) empirically better performance in

most cases. The expected tail risk minimization preserves a particular property that minimizing proportional worst-case fast adaptation seldom sacrifices the average performance. These advantages are also why we call our approach *simple yet effective*.

**Comparison with Other Heuristic Optimization Strategies.** To sum up the proposed optimization strategy, we empirically highlight the following points regarding the adopted heuristic strategies. There exist a couple of approximate algorithms for CVaR$_\alpha$ optimization. In comparison, the crude Monte Carlo one is the simplest for VaR$_\alpha$ estimates. It has an improvement guarantee under certain conditions, leaving it easier to analyze. Besides, we also compare ours to the risk reweighted method (Sagawa et al., 2020) in previously used benchmarks, and please take a closer look at that in Section (K).

# B   Pseudo Algorithms of DR-MAML & DR-CNPs

---

**Algorithm 1:** DR-MAML

---

**Input**  : Task distribution $p(\tau)$; Confidence level $\alpha$; Task batch size $\mathcal{B}$; Learning rates: $\lambda_1$ and $\lambda_2$.
**Output**: Meta-trained model parameter $\vartheta$.
Randomly initialize the model parameter $\vartheta$;
**while** *not converged* **do**

    Sample a batch of tasks $\{\tau_i\}_{i=1}^{\mathcal{B}} \sim p(\tau)$;
    // inner loop via gradient descent
    **for** $i = 1$ *to* $\mathcal{B}$ **do**

        Evaluate the gradient: $\nabla_\vartheta \ell(\mathfrak{D}_{\tau_i}^C; \vartheta)$ in Eq. (7);
        Perform task-specific gradient updates:
        $\vartheta_i \leftarrow \vartheta - \lambda_1 \nabla_\vartheta \ell(\mathfrak{D}_{\tau_i}^C; \vartheta)$;

    **end**

    // estimate VaR$_\alpha[\ell(\mathcal{T}, \vartheta)] \approx \hat{\xi}_\alpha$
    Evaluate performance $\mathcal{L}_\mathcal{B} = \{\ell(\mathfrak{D}_{\tau_i}^T; \vartheta_i)\}_{i=1}^{\mathcal{B}}$;
    Estimate VaR$_\alpha[\ell(\mathcal{T}, \vartheta)]$ and set $\xi = \hat{\xi}_\alpha$ in Eq. (7) with either percentile rank or density estimators;
    // outer loop via gradient descent
    Screen the subset $\mathcal{L}_{\hat{\mathcal{B}}} = \{\ell(\mathfrak{D}_{\hat{\tau}_i}^T; \vartheta_i)\}_{i=1}^K$ with $\hat{\xi}_\alpha$ for meta initialization updates;
    $\vartheta \leftarrow \vartheta - \lambda_2 \nabla_\vartheta \sum_{i=1}^K \ell(\mathfrak{D}_{\hat{\tau}_i}^T; \vartheta_i)$ in Eq. (7);
**end**

---

---

**Algorithm 2:** DR-CNP

---

**Input**  : Task distribution $p(\tau)$; Confidence level $\alpha$; Task batch size $\mathcal{B}$; Learning rate $\lambda$.
**Output**: Meta-trained model parameter $\vartheta$.
Randomly initialize the model parameter $\vartheta$;
**while** *not converged* **do**

    Sample a batch of tasks $\{\tau_i\}_{i=1}^{\mathcal{B}} \sim p(\tau)$;
    // estimate VaR$_\alpha[\ell(\mathcal{T}, \vartheta)]$
    Evaluate performance $\mathcal{L}_\mathcal{B} = \{\ell(\mathfrak{D}_{\tau_i}^T; z, \vartheta_i)\}_{i=1}^{\mathcal{B}}$;
    Estimate VaR$_\alpha[\ell(\mathcal{T}, \vartheta)] \approx \hat{\xi}_\alpha$ with either percentile rank or density estimators;
    // execute gradient descent
    Screen the subset $\mathcal{L}_{\hat{\mathcal{B}}} = \{\ell(\mathfrak{D}_{\hat{\tau}_i}^T; z, \vartheta)\}_{i=1}^K$ with $\hat{\xi}_\alpha$ for meta initialization updates;
    $\vartheta \leftarrow \vartheta - \lambda \nabla_\vartheta \sum_{i=1}^K \ell(\mathfrak{D}_{\hat{\tau}_i}^T; z, \vartheta)$ in Eq. (8);
**end**

---

# C  Properties of Risk Minimization Principles

## C.1  Stochastic Optimization in the Constrained Form

In the section above, meta learning optimization objectives are discussed within three different principles, respectively the average-case in Eq. (1), the worst-case in Eq. (2) and CVaR$_\alpha$ worst-case in Eq. (5). This subsection continues this topic and introduces the relaxation variable $\xi$ in the optimization objective. In this way, the robust fast adaptation can be reframed in the case of stochastic optimization with the probabilistic constraint.

We can equivalently express the minimization of the worst-case problem (Shalev-Shwartz and Wexler, 2016) within the following constrained stochastic optimization framework as follows.

$$\min_{\vartheta \in \Theta, \xi \in \mathbb{R}^+} \xi$$
$$\text{s.t. } \ell(\mathfrak{D}_\tau^T, \mathfrak{D}_\tau^C; \vartheta) \leq \xi \quad \text{with } \tau \in \mathcal{T} \tag{10}$$

## C.2  SGD Intractability of Meta Learning CVaR$_\alpha$ Optimization Objective

This subsection shows that directly stochastic gradient descent is intractable for meta learning to optimize CVaR$_\alpha$. Note that the normalized density distribution function $p_\alpha(\tau; \vartheta)$ implicitly depends on $\vartheta$ and $\alpha$, so we cannot access the exact form of such a distribution.

$$\nabla_\vartheta \mathcal{E}_\alpha(\vartheta) = \int p_\alpha(\tau; \vartheta) \Big[ \ell(\mathfrak{D}_\tau^T, \mathfrak{D}_\tau^C; \vartheta) \nabla_\vartheta \ln p_\alpha(\tau; \vartheta) + \nabla_\vartheta \ell(\mathfrak{D}_\tau^T, \mathfrak{D}_\tau^C; \vartheta) \Big] d\tau \tag{11a}$$

$$\approx \frac{1}{K} \sum_{k=1}^K \Big[ \underbrace{\ell(\mathfrak{D}_{\tau_k}^T, \mathfrak{D}_{\tau_k}^C; \vartheta) \nabla_\vartheta \ln p_\alpha(\tau_k; \vartheta)}_{\text{Score Function}} + \nabla_\vartheta \ell(\mathfrak{D}_{\tau_k}^T, \mathfrak{D}_{\tau_k}^C; \vartheta) \Big] \tag{11b}$$

As illustrated in Eq. (11), the stochastic gradient estimate is not plausible since $p_\alpha(\tau; \vartheta)$ has no closed form. Hence, heuristic or convex programming algorithms for specific cases are mostly used as this domain's optimization strategy. However, designing optimization strategies for non-convex cases is non-trivial in this domain and requires more consideration of theoretical guarantees.

## C.3  Risk-Sensitive Applications & Optimization Strategies

**Related Applications.** There are a number of applications concerning robust optimization with probabilistic constraints. Most of them are for the sake of safety. The risk principle CVaR$_\alpha$ firstly occurs in the financial domain as a coherent principle (Rockafellar et al., 2000). It enjoys much popularity in portfolio optimization (Quaranta and Zaffaroni, 2008). Gagne and Dayan (2021) adopt CVaR$_\alpha$ principles to improve the distributional reinforcement learning performance. To robustify robotic control, Pinto et al. (2017) varies hyper-parameters of Markov decision processes and optimizes proportional worst-case trajectories in policy optimization within the principle of CVaR$_\alpha$. In work (Wilder, 2018), a CVaR$_\alpha$ related strategy is devised to solve submodular optimization problems. Such a risk measure is also included in producing robust options (Hiraoka et al., 2019). *Regarding probabilistic robust meta learning within CVaR$_\alpha$ principles, there exists scarce related work until now.*

**Optimization Strategies.** Concerning the constrained stochastic optimization problem, we particularly overview related work in this subsection. In the past few decades, there emerge substantial optimization strategies together with theoretical analysis for convex risk functions in CVaR$_\alpha$ optimization (Nemirovski et al., 2009; Fan et al., 2017; Meng et al., 2020; Levy et al., 2020; Duchi and Namkoong, 2021; Wang et al., 2022). As for the min-max risk minimization principle, which focuses on the worst case instead of a propositional worst cases, some researchers have designed relaxation or other heuristic algorithms to handle convex or non-convex risk functions (Chen et al., 2017; Collins et al., 2020; Jiang et al., 2021; Hsieh et al., 2021). Nevertheless, it remains challenging to design algorithms with convergence guarantees for non-convex risk function cases. One of the latest CVaR$_\alpha$ work on non-convex risk functions is (Sagawa et al., 2020), where a risk reweighted algorithm for robust neural networks is proposed to handle distributional shifts. Moreover, most CVaR$_\alpha$ optimization in non-convex risk functions follows this type of risk reweighted strategies in applications.

**Remarks on Literature Work:** Risk functions for robust fast adaptation cases are mostly non-convex, and it is non-trivial to design optimization strategies with a convergence guarantee. Meanwhile, we also conduct comparison experiments between work in (Sagawa et al., 2020) and ours in meta learning downstream tasks. We refer the reader to Appendix (K) for more details and analysis.

## D    Computational Complexity

The primary difference between distributionally robust meta learning and expected risk based meta learning lies in that a fixed probabilistic portion of task gradients are considered in meta updates. Such an operation drives the optimization procedure to focus more on proportional vulnerable scenarios, increasing the robustness in worst cases.

However, the iteration of the surrogate function $\varphi(\hat{\xi}_{\alpha_t}; \vartheta)$ involves the screening of proportional worst cases since our strategies require extra evaluation of fast adaptation. This brings additional computational cost with complexity $\mathcal{O}\big(\mathcal{B}\log(\mathcal{B})\big)$, where $\mathcal{B}$ is the number of tasks in each batch. On the other hand, the proposed strategy performs sub-gradient updates instead of complete gradient updates. This helps reduce computational cost with complexity $\mathcal{O}\big(\alpha\mathcal{B}|\mathcal{M}|\big)$ in each iteration, where $|\mathcal{M}|$ corresponds to the scale of parameters of meta learning models and $\alpha$ is the confidence level in $\text{CVaR}_\alpha$ optimization.

Universally analyzing the computational complexity in meta learning is intractable since various meta learning methods exist. Some are gradient-based ones, while some are non-parametric ones. The exact number of iterations for convergence heavily relies on specific methods. In summary, given the same number of iterations and a fixed confidence level, the computational complexity difference for $\text{CVaR}_\alpha$ optimization in meta learning scenarios is $\mathcal{O}\big(\mathcal{B}(\alpha|\mathcal{M}| - \log(\mathcal{B}))\big)$.

To deepen understanding of our method, we explain more via specific examples. The main idea is to execute the sub-gradient operation over the batch of task gradients. Here we take the DR-MAML in Example (1) to show the operation and how the distributionally robust meta initialization is obtained:

$$\vartheta_{t+1}^{\text{meta}} \leftarrow \vartheta_t^{\text{meta}} - \lambda_1 \Big[ \sum_{i=1}^{\mathcal{B}} \nabla_\vartheta [\delta(\tau_i) \cdot \ell(\mathfrak{D}_{\tau_i}^T; \vartheta_t^{\tau_i})] \Big],$$
$$\text{with } \vartheta_t^{\tau_i} = \vartheta_t^{\text{meta}} - \lambda_2 \nabla_\vartheta \ell(\mathfrak{D}_{\tau_i}^C; \vartheta), \; \tau_i \sim p(\tau), \tag{12}$$

where $\lambda_1$ is the outer loop learning rate, $\lambda_2$ is the inner loop learning rate, and $\delta(\tau_i)$ is the indicator variable. Here $\delta(\tau_i) = 1$ in the case when the $\ell(\mathfrak{D}_{\tau_i}^T; \vartheta_t^{\tau_i})$ falls into the $(1 - \alpha)$-probabilistic worst-case region otherwise $\delta(\tau_i) = 0$.

As for the optimal rate for convergence or the generalization bound, it is up to specific meta learning methods and risk minimization principles. For worst-case risk minimization for meta learning, there already exists theoretical analysis in previous work, *e.g.*, optimization-based meta learning (Collins et al., 2020) when fast adaptation functions hold the convexity property. When it comes to more universal cases, considering diverse meta learning methods and optimization strategies, it is still challenging to estimate the optimal rate for convergence. Also, our considered scenarios exist distributional drift between meta training and meta testing task distributions, which makes it tough to derive the generalization bound.

Finally, note that our developed optimization strategies for meta learning are regardless of meta learning methods, and DR-MAMLs and DR-CNPs are merely two examples. For the sake of convenience, we only implement DR-MAML to compare with TR-MAML in this work.

## E    Proof of Remark (2)

***Remark (2).*** *If $\ell(\mathfrak{D}_\tau^T, \mathfrak{D}_\tau^C; \vartheta)$ is convex w.r.t. $\vartheta$, then Eq.s (5)/(6) are also convex functions. In this case, the optimization objective Eq. (6) of our interest can be resolved with the help of convex programming algorithms (Fan et al., 2017; Meng et al., 2020; Levy et al., 2020).*

**Proof:** We at first show that $\left[\ell(\mathfrak{D}_\tau^T, \mathfrak{D}_\tau^C; \vartheta) - \xi\right]^+ := \max\{\ell(\mathfrak{D}_\tau^T, \mathfrak{D}_\tau^C; \vartheta) - \xi, 0\}$ is convex *w.r.t* $\vartheta$ if $\ell(\mathfrak{D}_\tau^T, \mathfrak{D}_\tau^C; \vartheta)$ is convex *w.r.t.* $\vartheta$: For ease of derivation, let us redenote two functions $f_1(\xi; \vartheta) := \ell(\mathfrak{D}_\tau^T, \mathfrak{D}_\tau^C; \vartheta) - \xi$ and $f_2(\xi; \vartheta) := 0$.

With any constant $\lambda \in [0, 1]$ and any two parameters $\vartheta_1 \in \Theta$ and $\vartheta_2 \in \Theta$, we can have:

$$\left[\ell(\mathfrak{D}_\tau^T, \mathfrak{D}_\tau^C; \lambda\vartheta_1 + (1-\lambda)\vartheta_2) - \xi\right]^+ = f_i(\xi; \lambda\vartheta_1 + (1-\lambda)\vartheta_2) \text{ [for some } i \in \{1,2\}] \quad (13\text{a})$$

$$\leq \lambda f_i(\xi; \vartheta_1) + (1-\lambda)f_i(\xi; \vartheta_2) \quad (13\text{b})$$

$$\leq \lambda \max\{f_i(\xi; \vartheta_1)\}_{i=1,2} + (1-\lambda)\max\{f_i(\xi; \vartheta_2)\}_{i=1,2} \quad (13\text{c})$$

$$\leq \lambda\left[\ell(\mathfrak{D}_\tau^T, \mathfrak{D}_\tau^C; \vartheta_1) - \xi\right]^+ + (1-\lambda)\left[\ell(\mathfrak{D}_\tau^T, \mathfrak{D}_\tau^C; \vartheta_2) - \xi\right]^+, \quad (13\text{d})$$

which shows that the risk function with slack variables is convex *w.r.t.* $\vartheta$.

As $\varphi_\alpha(\xi; \vartheta) = \xi + \frac{1}{1-\alpha}\mathbb{E}_{p(\tau)}\left[\left[\ell(\mathfrak{D}_\tau^T, \mathfrak{D}_\tau^C; \vartheta) - \xi\right]^+\right]$ is the convex combination of the above mentioned convex function, the resulting $\varphi_\alpha(\xi; \vartheta)$ is naturally convex *w.r.t.* $\vartheta$.

# F  Proof of Proposition (1)

**Assumption (1):** *The meta risk function $\ell(\mathfrak{D}_\tau^T, \mathfrak{D}_\tau^C; \vartheta)$ is $\beta_\tau$-Lipschitz continuous w.r.t. $\vartheta$, which suggests: there exists a positive constant $\beta_\tau$ such that $\forall\{\vartheta, \vartheta'\} \in \Theta$:*

$$|\ell(\mathfrak{D}_\tau^T, \mathfrak{D}_\tau^C; \vartheta) - \ell(\mathfrak{D}_\tau^T, \mathfrak{D}_\tau^C; \vartheta')| \leq \beta_\tau||\vartheta - \vartheta'||.$$

**Assumption (2):** *For meta risk function values, the risk cumulative distribution function $F_\ell(l; \vartheta)$ is $\beta_\ell$-Lipschitz continuous w.r.t. $l$, and the implicit normalized probability density function of tasks $p_\alpha(\tau; \vartheta)$ is $\beta_\theta$-Lipschitz continuous w.r.t. $\vartheta$.*

**Assumption (3):** *For any valid $\vartheta \in \Theta$ and corresponding implicit normalized probability density function of tasks $p_\alpha(\tau; \vartheta)$, the meta risk function value can be bounded by a positive constant $\mathcal{L}_{\max}$:*

$$\sup_{\tau \in \Omega_\tau^\alpha} \ell(\mathfrak{D}_{\tau_i}^T, \mathfrak{D}_{\tau_i}^C; \vartheta) \leq \mathcal{L}_{\max}.$$

**Proposition (1):** *Under Assumptions (1)/(2)/(3), the meta learning optimization objective $\mathcal{E}_\alpha(\vartheta)$ in Eq. (3) is continuous w.r.t. $\vartheta$.*

**Proof:** Suppose that $\forall\{\vartheta, \vartheta'\} \in \Theta$ and $||\vartheta - \vartheta'|| < \delta$, we can have the following inequality based on Assumption (2):

$$\left|p_\alpha(\tau; \vartheta) - p_\alpha(\tau; \vartheta')\right| \leq \beta_\theta||\vartheta - \vartheta'||.$$

Meanwhile, we can have the following inequality based on Assumption (1):

$$\left|\ell(\mathfrak{D}_\tau^T, \mathfrak{D}_\tau^C; \vartheta) - \ell(\mathfrak{D}_\tau^T, \mathfrak{D}_\tau^C; \vartheta')\right| \leq \beta_\tau||\vartheta - \vartheta'||.$$

Together with the boundness Assumption (3) of meta risk function value:

$$\sup_{\tau \in \Omega_\tau^\alpha} \ell(\mathfrak{D}_{\tau_i}^T, \mathfrak{D}_{\tau_i}^C; \vartheta) \leq \mathcal{L}_{\max},$$

we can roughly estimate the probabilistic constrained expected meta risk function values as follows:

$$\left|\mathcal{E}_\alpha(\vartheta) - \mathcal{E}_\alpha(\vartheta')\right| = \left|\mathbb{E}_{p_\alpha(\tau;\vartheta)}\left[\ell(\mathfrak{D}_\tau^T, \mathfrak{D}_\tau^C; \vartheta)\right] - \mathbb{E}_{p_\alpha(\tau;\vartheta')}\left[\ell(\mathfrak{D}_\tau^T, \mathfrak{D}_\tau^C; \vartheta')\right]\right| \tag{14a}$$

$$= \left|\mathbb{E}_{p_\alpha(\tau;\vartheta)}\left[\ell(\mathfrak{D}_\tau^T, \mathfrak{D}_\tau^C; \vartheta)\right] - \mathbb{E}_{p_\alpha(\tau;\vartheta')}\left[\ell(\mathfrak{D}_\tau^T, \mathfrak{D}_\tau^C; \vartheta)\right]\right. \tag{14b}$$

$$\left. + \mathbb{E}_{p_\alpha(\tau;\vartheta')}\left[\ell(\mathfrak{D}_\tau^T, \mathfrak{D}_\tau^C; \vartheta) - \ell(\mathfrak{D}_\tau^T, \mathfrak{D}_\tau^C; \vartheta')\right]\right| \tag{14c}$$

$$\leq \int \left|p_\alpha(\tau;\vartheta) - p_\alpha(\tau;\vartheta')\right|\ell(\mathfrak{D}_\tau^T, \mathfrak{D}_\tau^C; \vartheta)d\tau + \mathbb{E}_{p_\alpha(\tau;\vartheta')}\left[\left|\ell(\mathfrak{D}_\tau^T, \mathfrak{D}_\tau^C; \vartheta) - \ell(\mathfrak{D}_\tau^T, \mathfrak{D}_\tau^C; \vartheta')\right|\right] \tag{14d}$$

$$\leq \beta_\theta ||\vartheta - \vartheta'|| \sup_{\tau \in \Omega_\tau^\alpha}\{\ell(\mathfrak{D}_\tau^T, \mathfrak{D}_\tau^C; \vartheta)\} + \sup_{\tau \in \Omega_\tau^\alpha}\{\beta_\tau\}||\vartheta - \vartheta'|| \tag{14e}$$

$$= \left(\beta_\theta \mathcal{L}_{\max} + \beta_{\max}\right)||\vartheta - \vartheta'||. \tag{14f}$$

From the above inequality, we can see that $\mathcal{E}_\alpha(\vartheta) - \mathcal{E}_\alpha(\vartheta')$ is a $(\beta_\theta\mathcal{L}_{\max} + \beta_{\max})$–Lipschitz continuous *w.r.t.* $\vartheta$. As a result, we demonstrate Proposition (1).

## G   Proof of Proposition (2)

***Proposition (2):*** *Suppose there exists $\delta \in \mathbb{R}^+$ such that $|\xi_\alpha(\vartheta) - \hat{\xi}_\alpha(\vartheta)| < \delta$ with $\hat{\xi}_\alpha(\vartheta)$ an estimate of $\xi_\alpha(\vartheta)$. Then there exists a constant $\kappa_\alpha = \max\{\frac{2-\alpha}{1-\alpha}, \frac{\alpha}{1-\alpha}\}$ such that*

$$\varphi_\alpha(\hat{\xi}_\alpha(\vartheta); \vartheta) - \kappa_\alpha\delta < \mathcal{E}_\alpha(\vartheta) \leq \varphi_\alpha(\hat{\xi}_\alpha(\vartheta); \vartheta).$$

***Proof:*** Based on the direct deduction in work (Rockafellar et al., 2000), we know that for any $\vartheta \in \Theta$, the inequality holds: $\varphi_\alpha(\xi_\alpha; \vartheta) \leq \varphi_\alpha(\hat{\xi}_\alpha; \vartheta)$.

In the case when $\hat{\xi}_\alpha = \xi_\alpha + \delta$ with $\delta \in \mathbb{R}^+$, the probability space of the task $\Omega_\tau$ can be respectively partitioned into three disjoint probability space:

$$\mathbb{P}^-(\tau) = \mathbb{P}\left(\{\mathcal{M}_\vartheta^{-1}(\ell)|\ell(\mathfrak{D}_\tau^T, \mathfrak{D}_\tau^C; \vartheta) \leq \xi_\alpha, \tau \in \Omega_\tau\}\right) \tag{15a}$$

$$\mathbb{P}^+(\tau) = \mathbb{P}\left(\{\mathcal{M}_\vartheta^{-1}(\ell)|\xi_\alpha < \ell(\mathfrak{D}_\tau^T, \mathfrak{D}_\tau^C; \vartheta) \leq \xi_\alpha + \delta, \tau \in \Omega_\tau\}\right) \tag{15b}$$

$$\mathbb{P}^{++}(\tau) = \mathbb{P}\left(\{\mathcal{M}_\vartheta^{-1}(\ell)|\ell(\mathfrak{D}_\tau^T, \mathfrak{D}_\tau^C; \vartheta) > \xi_\alpha + \delta, \tau \in \Omega_\tau\}\right). \tag{15c}$$

As a result, we can estimate the difference between the two terms as follows:

$$\varphi_\alpha(\hat{\xi}_\alpha; \vartheta) - \varphi_\alpha(\xi_\alpha; \vartheta) = \hat{\xi}_\alpha - \xi_\alpha \tag{16a}$$

$$+ \frac{1}{1-\alpha}\mathbb{E}_{\tau \sim \mathbb{P}^+(\tau)}\left[\ell(\mathfrak{D}_\tau^T, \mathfrak{D}_\tau^C; \vartheta) - \xi_\alpha\right] + \frac{1}{1-\alpha}\mathbb{E}_{\tau \sim \mathbb{P}^{++}(\tau)}\left[\xi_\alpha - \hat{\xi}_\alpha\right] \tag{16b}$$

$$\leq \delta + \frac{1}{1-\alpha}\mathbb{E}_{\tau \sim \mathbb{P}^+(\tau)}\left[\delta\right] - \frac{1}{1-\alpha}\mathbb{E}_{\tau \sim \mathbb{P}^{++}(\tau)}\left[\delta\right] \leq \delta + \frac{1}{1-\alpha}\delta = \frac{2-\alpha}{1-\alpha}\delta. \tag{16c}$$

Similarly, in the case when $\hat{\xi}_\alpha = \xi_\alpha - \delta$ with $\delta \in \mathbb{R}^+$, the probability space of the task $\Omega_\tau$ can be partitioned into three disjoint space with the probability respectively:

$$\mathbb{P}^{--}(\tau) = \mathbb{P}\Big(\{\mathcal{M}_\vartheta^{-1}(\ell)|\ell(\mathfrak{D}_\tau^T, \mathfrak{D}_\tau^C; \vartheta) \le \xi_\alpha - \delta, \tau \in \Omega_\tau\}\Big) \tag{17a}$$

$$\mathbb{P}^-(\tau) = \mathbb{P}\Big(\{\mathcal{M}_\vartheta^{-1}(\ell)| - \delta < \ell(\mathfrak{D}_\tau^T, \mathfrak{D}_\tau^C; \vartheta) - \xi_\alpha < 0, \tau \in \Omega_\tau\}\Big) \tag{17b}$$

$$\mathbb{P}^+(\tau) = \mathbb{P}\Big(\{\mathcal{M}_\vartheta^{-1}(\ell)|\ell(\mathfrak{D}_\tau^T, \mathfrak{D}_\tau^C; \vartheta) \ge \xi_\alpha, \tau \in \Omega_\tau\}\Big). \tag{17c}$$

As a result, we can estimate the difference between the two terms as follows:

$$\varphi_\alpha(\hat{\xi}_\alpha; \vartheta) - \varphi_\alpha(\xi_\alpha; \vartheta) = \hat{\xi}_\alpha - \xi_\alpha \tag{18a}$$

$$+ \frac{1}{1-\alpha}\mathbb{E}_{\tau \sim \mathbb{P}^-(\tau)}\Big[\ell(\mathfrak{D}_\tau^T, \mathfrak{D}_\tau^C; \vartheta) - \hat{\xi}_\alpha\Big] + \frac{1}{1-\alpha}\mathbb{E}_{\tau \sim \mathbb{P}^+(\tau)}\Big[\xi_\alpha - \hat{\xi}_\alpha\Big] \tag{18b}$$

$$\le -\delta + \frac{1}{1-\alpha}\mathbb{E}_{\tau \sim \mathbb{P}^-(\tau)}\Big[\delta\Big] + \frac{1}{1-\alpha}\mathbb{E}_{\tau \sim \mathbb{P}^+(\tau)}\Big[\delta\Big] \le -\delta + \frac{1}{1-\alpha}\delta = \frac{\alpha}{1-\alpha}\delta. \tag{18c}$$

Based on the inequalities (16)/(18), we can have $\kappa_\alpha = \max\{\frac{2-\alpha}{1-\alpha}, \frac{\alpha}{1-\alpha}\}$ such that the proposition is verified as $\varphi_\alpha(\hat{\xi}_\alpha; \vartheta) - \kappa_\alpha \delta < \varphi_\alpha(\xi_\alpha; \vartheta)$.

## H Proof of Improvement Guarantee in Theorem (1)

***Theorem (1):*** *Under assumptions (1)/(2)/(3), suppose that the estimate error with the crude Monte Carlo holds: $|\hat{\xi}_{\alpha_t} - \xi_{\alpha_t}| \le \frac{\lambda}{\beta_\ell(1-\alpha)^2}, \forall t \in \mathbb{N}^+$, with the subscript $t$ the iteration number, $\lambda$ the learning rate in stochastic gradient descent, $\beta_\ell$ the Lipschitz constant of the risk cumulative distribution function, and $\alpha$ the confidence level. Then the proposed heuristic algorithm with the crude Monte Carlo can produce at least a local optimum for distributionally robust fast adaptation.*

***Proof:*** In the main paper, Fig. (2) provides the outline of the improvement guarantee proof. Note that $\hat{\xi}_{\alpha_t}$ is an estimate of $\xi_{\alpha_t}$ with the help of Monte Carlo samples, and this depends on the model parameters $\vartheta_t$ in optimization. Performing the gradient updates *w.r.t.* the surrogate function $\varphi_\alpha(\hat{\xi}_\alpha; \vartheta)$, we can have the following equation with a small step-size learning rate $\lambda$:

$$\text{Gradient Descent}: \vartheta_{t+1} = \vartheta_t - \lambda\nabla_\vartheta\varphi_\alpha(\hat{\xi}_\alpha; \vartheta)$$
$$\Rightarrow \text{Monotonic Sequence}: \varphi_\alpha(\hat{\xi}_{\alpha_t}; \vartheta_{t+1}) \le \varphi_\alpha(\hat{\xi}_{\alpha_t}; \vartheta_t). \tag{19}$$

To verify the improvement guarantee, we need to show that with the meta model parameters derived from the surrogate function:

$$\varphi_\alpha(\xi_{\alpha_{t+1}}; \vartheta_{t+1}) \le \varphi_\alpha(\xi_{\alpha_t}; \vartheta_t). \tag{20}$$

With the previous deduction $\varphi_\alpha(\xi_{\alpha_{t+1}}; \vartheta_{t+1}) \le \varphi_\alpha(\xi_{\alpha_t}; \vartheta_{t+1})$ from the property of CVaR$_\alpha$, the demonstration is equivalently reduced to show that:

$$\varphi_\alpha(\xi_{\alpha_t}; \vartheta_{t+1}) \le \varphi_\alpha(\xi_{\alpha_t}; \vartheta_t). \tag{21}$$

We can perform the one order Taylor expansion with Peano's form of remainders *w.r.t.* $\varphi_\alpha(\xi_{\alpha_t}; \vartheta)$ around the point $\vartheta_t$:

$$\varphi_\alpha(\xi_{\alpha_t}; \vartheta_{t+1}) = \varphi_\alpha(\xi_{\alpha_t}; \vartheta_t) - \lambda\Big[\nabla_\vartheta\varphi_\alpha(\hat{\xi}_{\alpha_t}; \vartheta)|_{\vartheta=\vartheta_t}^T \cdot \nabla_\vartheta\varphi_\alpha(\xi_{\alpha_t}; \vartheta)|_{\vartheta=\vartheta_t}\Big]$$
$$+ \mathcal{O}(|\vartheta_{t+1} - \vartheta_t|) \le \varphi_\alpha(\xi_{\alpha_t}; \vartheta_t). \tag{22}$$

In the case when $\hat{\xi}_\alpha = \xi_\alpha + \delta$ with $\delta \in \mathbb{R}^+$, we use the partitioned task probability space in Eq. (15) and can derive the gradient estimate:

$$\nabla_\vartheta\varphi_\alpha(\hat{\xi}_{\alpha_t}; \vartheta)|_{\vartheta=\vartheta_t}^T = \frac{1}{1-\alpha}\Big[\mathbb{E}_{\tau \sim \mathbb{P}^+(\tau)}\Big[\nabla_\vartheta\ell(\mathfrak{D}_\tau^T, \mathfrak{D}_\tau^C; \vartheta)|_{\vartheta=\vartheta_t}\Big]\Big]$$
$$+ \frac{1}{1-\alpha}\Big[\mathbb{E}_{\tau \sim \mathbb{P}^{++}(\tau)}\Big[\nabla_\vartheta\ell(\mathfrak{D}_\tau^T, \mathfrak{D}_\tau^C; \vartheta)|_{\vartheta=\vartheta_t}\Big]\Big]. \tag{23}$$

With $|F_\ell(\hat{\xi}_\alpha; \vartheta) - F_\ell(\xi_\alpha; \vartheta)| \leq \beta_\ell \delta$ in the Assumption (2),

$$-\mathbb{E}_{\tau \sim \mathbb{P}^+(\tau)}\left[\nabla_\vartheta \ell(\mathfrak{D}_\tau^T, \mathfrak{D}_\tau^C; \vartheta)|_{\vartheta=\vartheta_t}\right]^T \cdot \mathbb{E}_{\tau \sim \mathbb{P}^{++}(\tau)}\left[\nabla_\vartheta \ell(\mathfrak{D}_\tau^T, \mathfrak{D}_\tau^C; \vartheta)|_{\vartheta=\vartheta_t}\right]$$
(24a)

$$\leq ||\mathbb{E}_{\tau \sim \mathbb{P}^+(\tau)}\left[\nabla_\vartheta \ell(\mathfrak{D}_\tau^T, \mathfrak{D}_\tau^C; \vartheta)|_{\vartheta=\vartheta_t}\right]|| \cdot ||\mathbb{E}_{\tau \sim \mathbb{P}^{++}(\tau)}\left[\nabla_\vartheta \ell(\mathfrak{D}_\tau^T, \mathfrak{D}_\tau^C; \vartheta)|_{\vartheta=\vartheta_t}\right]||$$
(24b)

$$\leq \mathbb{E}_{\tau \sim \mathbb{P}^+(\tau)}\left[\sup_{\tau \in \Omega_\tau} ||\nabla_\vartheta \ell(\mathfrak{D}_\tau^T, \mathfrak{D}_\tau^C; \vartheta)|_{\vartheta=\vartheta_t}||\right] \cdot \mathbb{E}_{\tau \sim \mathbb{P}^{++}(\tau)}\left[\sup_{\tau \in \Omega_\tau} ||\nabla_\vartheta \ell(\mathfrak{D}_\tau^T, \mathfrak{D}_\tau^C; \vartheta)|_{\vartheta=\vartheta_t}||\right]$$
(24c)

$$\leq \beta_\ell \delta (1-\alpha) \left(\sup_{\tau \in \Omega_\tau} ||\nabla_\vartheta \ell(\mathfrak{D}_\tau^T, \mathfrak{D}_\tau^C; \vartheta)|_{\vartheta=\vartheta_t}||\right)^2 = \beta_\ell \delta (1-\alpha)\mu^2,$$
(24d)

where $\mu$ defines the $\sup_{\tau \in \Omega_\tau} ||\nabla_\vartheta \ell(\mathfrak{D}_\tau^T, \mathfrak{D}_\tau^C; \vartheta)|_{\vartheta=\vartheta_t}||$.

Then we can derive the following inequality with the deduction from Eq. (24):

$$\varphi_\alpha(\xi_{\alpha_t}; \vartheta_{t+1}) = \varphi_\alpha(\xi_{\alpha_t}; \vartheta_t) \quad \text{(25a)}$$

$$-\lambda\left[\nabla_\vartheta \varphi_\alpha(\hat{\xi}_{\alpha_t}; \vartheta)|_{\vartheta=\vartheta_t}^T \cdot \nabla_\vartheta \varphi_\alpha(\xi_{\alpha_t}; \vartheta)|_{\vartheta=\vartheta_t}\right] + \mathcal{O}(|\vartheta_{t+1} - \vartheta_t|) \quad \text{(25b)}$$

$$= \varphi_\alpha(\xi_{\alpha_t}; \vartheta_t) \quad \text{(25c)}$$

$$-\frac{\lambda}{(1-\alpha)^2}\left[\mathbb{E}_{\tau \sim \mathbb{P}^{++}(\tau)}\left[\nabla_\vartheta \ell(\mathfrak{D}_\tau^T, \mathfrak{D}_\tau^C; \vartheta)|_{\vartheta=\vartheta_t}\right]^T \cdot \mathbb{E}_{\tau \sim \mathbb{P}^{++}(\tau)}\left[\nabla_\vartheta \ell(\mathfrak{D}_\tau^T, \mathfrak{D}_\tau^C; \vartheta)|_{\vartheta=\vartheta_t}\right]\right] \quad \text{(25d)}$$

$$-\frac{\lambda}{(1-\alpha)^2}\left[\mathbb{E}_{\tau \sim \mathbb{P}^+(\tau)}\left[\nabla_\vartheta \ell(\mathfrak{D}_\tau^T, \mathfrak{D}_\tau^C; \vartheta)|_{\vartheta=\vartheta_t}\right]^T \cdot \mathbb{E}_{\tau \sim \mathbb{P}^{++}(\tau)}\left[\nabla_\vartheta \ell(\mathfrak{D}_\tau^T, \mathfrak{D}_\tau^C; \vartheta)|_{\vartheta=\vartheta_t}\right]\right] \quad \text{(25e)}$$

$$+\mathcal{O}(|\vartheta_{t+1} - \vartheta_t|) \quad \text{(25f)}$$

$$\leq \varphi_\alpha(\xi_{\alpha_t}; \vartheta_t) - \frac{\lambda||\nu_1||_2^2}{(1-\alpha)^2} + \beta_\ell \delta (1-\alpha)\mu^2 + \mathcal{O}(|\vartheta_{t+1} - \vartheta_t|), \quad \text{(25g)}$$

where $\nu_1 = \mathbb{E}_{\tau \sim \mathbb{P}^{++}(\tau)}\left[\nabla_\vartheta \ell(\mathfrak{D}_\tau^T, \mathfrak{D}_\tau^C; \vartheta)|_{\vartheta=\vartheta_t}\right]$.

To ensure the existence of improvement guarantee, we need that the following inequality holds:

$$\varphi_\alpha(\xi_{\alpha_t}; \vartheta_t) - \frac{\lambda||\nu_1||_2^2}{(1-\alpha)^2} + \beta_\ell \delta (1-\alpha)\mu^2 + \mathcal{O}(|\vartheta_{t+1} - \vartheta_t|) \leq \varphi_\alpha(\xi_{\alpha_t}; \vartheta_t).$$
(26)

Similarly, in the case when $\hat{\xi}_\alpha = \xi_\alpha - \delta$ with $\delta \in \mathbb{R}^+$, we use the probability space of partitioned tasks in Eq. (17) and can derive the gradient estimate:

$$\nabla_\vartheta \varphi_\alpha(\hat{\xi}_{\alpha_t}; \vartheta)|_{\vartheta=\vartheta_t}^T = \frac{1}{1-\alpha}\left[\mathbb{E}_{\tau \sim \mathbb{P}^-(\tau)}\left[\nabla_\vartheta \ell(\mathfrak{D}_\tau^T, \mathfrak{D}_\tau^C; \vartheta)|_{\vartheta=\vartheta_t}\right] + \mathbb{E}_{\tau \sim \mathbb{P}^+(\tau)}\left[\nabla_\vartheta \ell(\mathfrak{D}_\tau^T, \mathfrak{D}_\tau^C; \vartheta)|_{\vartheta=\vartheta_t}\right]\right].$$
(27)

With $|F_\ell(\hat{\xi}_\alpha; \vartheta) - F_\ell(\xi_\alpha; \vartheta)| \leq \beta_\ell \delta$ in the Assumption (2), the following formula can be easily verified:

$$-\mathbb{E}_{\tau \sim \mathbb{P}^-(\tau)}\left[\nabla_\vartheta \ell(\mathfrak{D}_\tau^T, \mathfrak{D}_\tau^C; \vartheta)|_{\vartheta=\vartheta_t}\right]^T \cdot \mathbb{E}_{\tau \sim \mathbb{P}^+(\tau)}\left[\nabla_\vartheta \ell(\mathfrak{D}_\tau^T, \mathfrak{D}_\tau^C; \vartheta)|_{\vartheta=\vartheta_t}\right]$$

(28a)

$$\leq ||\mathbb{E}_{\tau \sim \mathbb{P}^-(\tau)}\left[\nabla_\vartheta \ell(\mathfrak{D}_\tau^T, \mathfrak{D}_\tau^C; \vartheta)|_{\vartheta=\vartheta_t}\right]|| \cdot ||\mathbb{E}_{\tau \sim \mathbb{P}^+(\tau)}\left[\nabla_\vartheta \ell(\mathfrak{D}_\tau^T, \mathfrak{D}_\tau^C; \vartheta)|_{\vartheta=\vartheta_t}\right]||$$

(28b)

$$\leq \mathbb{E}_{\tau \sim \mathbb{P}^-(\tau)}\left[\sup_{\tau \in \Omega_\tau} ||\nabla_\vartheta \ell(\mathfrak{D}_\tau^T, \mathfrak{D}_\tau^C; \vartheta)|_{\vartheta=\vartheta_t}||\right] \cdot \mathbb{E}_{\tau \sim \mathbb{P}^+(\tau)}\left[\sup_{\tau \in \Omega_\tau} ||\nabla_\vartheta \ell(\mathfrak{D}_\tau^T, \mathfrak{D}_\tau^C; \vartheta)|_{\vartheta=\vartheta_t}||\right]$$

(28c)

$$\leq \beta_\ell \delta(1-\alpha)\left(\sup_{\tau \in \Omega_\tau} ||\nabla_\vartheta \ell(\mathfrak{D}_\tau^T, \mathfrak{D}_\tau^C; \vartheta)|_{\vartheta=\vartheta_t}||\right)^2 = \beta_\ell \delta(1-\alpha)\mu^2.$$

(28d)

Once again, we can derive the following inequality with the deduction from Eq. (28):

$$\varphi_\alpha(\xi_{\alpha_t}; \vartheta_{t+1}) = \varphi_\alpha(\xi_{\alpha_t}; \vartheta_t) - \lambda\left[\nabla_\vartheta \varphi_\alpha(\hat{\xi}_{\alpha_t}; \vartheta)|_{\vartheta=\vartheta_t}^T \cdot \nabla_\vartheta \varphi_\alpha(\xi_{\alpha_t}; \vartheta)|_{\vartheta=\vartheta_t}\right] + \mathcal{O}(|\vartheta_{t+1} - \vartheta_t|)$$

(29a)

$$= \varphi_\alpha(\xi_{\alpha_t}; \vartheta_t)$$

(29b)

$$-\frac{\lambda}{(1-\alpha)^2}\left[\mathbb{E}_{\tau \sim \mathbb{P}^+(\tau)}\left[\nabla_\vartheta \ell(\mathfrak{D}_\tau^T, \mathfrak{D}_\tau^C; \vartheta)|_{\vartheta=\vartheta_t}\right]^T \cdot \mathbb{E}_{\tau \sim \mathbb{P}^+(\tau)}\left[\nabla_\vartheta \ell(\mathfrak{D}_\tau^T, \mathfrak{D}_\tau^C; \vartheta)|_{\vartheta=\vartheta_t}\right]\right]$$

(29c)

$$-\frac{\lambda}{(1-\alpha)^2}\left[\mathbb{E}_{\tau \sim \mathbb{P}^-(\tau)}\left[\nabla_\vartheta \ell(\mathfrak{D}_\tau^T, \mathfrak{D}_\tau^C; \vartheta)|_{\vartheta=\vartheta_t}\right]^T \cdot \mathbb{E}_{\tau \sim \mathbb{P}^+(\tau)}\left[\nabla_\vartheta \ell(\mathfrak{D}_\tau^T, \mathfrak{D}_\tau^C; \vartheta)|_{\vartheta=\vartheta_t}\right]\right]$$

(29d)

$$+\mathcal{O}(|\vartheta_{t+1} - \vartheta_t|)$$

(29e)

$$\leq \varphi_\alpha(\xi_{\alpha_t}; \vartheta_t) - \frac{\lambda||\nu_2||_2^2}{(1-\alpha)^2} + \beta_\ell \delta(1-\alpha)\mu^2 + \mathcal{O}(|\vartheta_{t+1} - \vartheta_t|),$$

(29f)

where $\nu_2 = \mathbb{E}_{\tau \sim \mathbb{P}^+(\tau)}\left[\nabla_\vartheta \ell(\mathfrak{D}_\tau^T, \mathfrak{D}_\tau^C; \vartheta)|_{\vartheta=\vartheta_t}\right]$.

To ensure the existence of improvement guarantee, we need that the following holds:

$$\varphi_\alpha(\xi_{\alpha_t}; \vartheta_t) - \frac{\lambda||\nu_2||_2^2}{(1-\alpha)^2} + \beta_\ell \delta(1-\alpha)\mu^2 + \mathcal{O}(|\vartheta_{t+1} - \vartheta_t|) \leq \varphi_\alpha(\xi_{\alpha_t}; \vartheta_t).$$

(30)

Considering the above two cases and estimated bounds Eq. (26)/(30), we can roughly estimate the upper bound of the required $\delta$ to guarantee performance improvement using our developed strategy in optimization:

$$\delta \leq \frac{\lambda||\nu_m||_2^2}{\beta_\ell(1-\alpha)^3\mu^2},$$

(31)

where $\lambda$ is the formerly mentioned learning rate, $||\nu_m||_2^2$ is $\max\{||\nu_1||_2^2, ||\nu_2||_2^2\}$, and $\mu$ is supermum of the meta risk function derivatives in the task domain $\sup_{\tau \in \Omega_\tau} ||\nabla_\vartheta \ell(\mathfrak{D}_\tau^T, \mathfrak{D}_\tau^C; \vartheta)|_{\vartheta=\vartheta_t}||$.

With the help of Jensen inequality, $||\nu_i||_2^2$ can be roughly bounded as:

$$||\nu_i||_2^2 \leq \mathbb{E}_{\tau \sim \mathbb{P}^+(\tau)}\left[\left[\nabla_\vartheta \ell(\mathfrak{D}_\tau^T, \mathfrak{D}_\tau^C; \vartheta)|_{\vartheta=\vartheta_t}\right]^T \cdot \left[\nabla_\vartheta \ell(\mathfrak{D}_\tau^T, \mathfrak{D}_\tau^C; \vartheta)|_{\vartheta=\vartheta_t}\right]\right]$$

$$\leq (1-\alpha)\left(\sup_{\tau \in \Omega_\tau} ||\nabla_\vartheta \ell(\mathfrak{D}_\tau^T, \mathfrak{D}_\tau^C; \vartheta)|_{\vartheta=\vartheta_t}||\right)^2 = (1-\alpha)\mu^2.$$

(32)

With Eq.s (31)/(32), we can finally obtain the necessary condition for improvement guarantee:

$$\delta \leq \frac{\lambda}{\beta_\ell (1-\alpha)^2}. \tag{33}$$

# I  Proof of Approximation Error in Theorem (2)

This section is to build up connections between the number of Monte Carlo samples in estimating $\mathrm{VaR}_\alpha$ and the gap of solutions between the approximately derived one and the theoretical one.

**Theorem 2 (Gaps of Optimized Solutions)** *Suppose $F_\ell(l;\vartheta) \in \mathcal{C}^2$ in l-domain. With meta trained $\vartheta_*$, the crude Monte Carlo estimate of $\hat{\xi}_\alpha$, the constant $\kappa_\alpha$ in Proposition (2), and the sufficiently large number of the task batch $\mathcal{B}$, and $\mathcal{R}_\mathcal{B} = \mathcal{O}(\mathcal{B}^{-3/4} \ln \mathcal{B})$, we have the expected error between the exact optimum and the approximate optimum:*

$$\mathcal{E}_\alpha(\vartheta_*) \leq \varphi_\alpha(\hat{\xi}_\alpha; \vartheta_*) + \kappa_\alpha \left[ \frac{\alpha - \hat{F}_\ell(\hat{\xi}_\alpha; \mathcal{B}, \vartheta_*)}{\frac{dF_\ell(\xi;\vartheta)}{d\xi}|_{\xi=\xi_\alpha}} + \mathcal{R}_\mathcal{B} \right].$$

***Proof:*** As noted in (Bahadur, 1966), with assumptions that the cumulative distribution function $F_\ell(l;\vartheta)$'s second order derivative is continuous in $l$-domain, namely $F_\ell(l;\vartheta) \in \mathcal{C}^2$, and $\frac{dF_\ell(\xi;\vartheta)}{d\xi}|_{\xi=\xi_\alpha}$, the quantile estimate with crude Monte Carlo can be asymptotically written in the form with the help of central limit theory (Rosenblatt, 1956):

$$\hat{\xi}_\alpha - \xi_\alpha = \frac{\alpha - \hat{F}_\ell(\hat{\xi}_\alpha; \mathcal{B}, \vartheta_*)}{\frac{dF_\ell(\xi;\vartheta_*)}{d\xi}|_{\xi=\xi_\alpha}} + \mathcal{R}_\mathcal{B}, \quad \text{with } \mathcal{R}_\mathcal{B} = \mathcal{O}(\mathcal{B}^{-3/4} \ln \mathcal{B}) \text{ when } \mathcal{B} \to \infty, \tag{34}$$

where the empirical cumulative distribution function is computed as follows.

With the sampled meta risk function values $\mathcal{L}_\mathcal{B} = \{\ell(\mathfrak{D}_{\tau_i}^T, \mathfrak{D}_{\tau_i}^C; \vartheta)\}_{i=1}^\mathcal{B}$, we rank them by values as $\hat{\mathcal{L}}_\mathcal{B} = \{\ell(\mathfrak{D}_{\hat{\tau}_i}^T, \mathfrak{D}_{\hat{\tau}_i}^C; \vartheta)\}_{i=1}^\mathcal{B}$, which means $\ell(\mathfrak{D}_{\hat{\tau}_{i-1}}^T, \mathfrak{D}_{\hat{\tau}_{i-1}}^C; \vartheta) \leq \ell(\mathfrak{D}_{\hat{\tau}_i}^T, \mathfrak{D}_{\hat{\tau}_i}^C; \vartheta)$. Then the empirical cumulative distribution function with these order statistics (Barabás, 1987) can be written as:

$$\hat{F}_\ell(\xi; \mathcal{B}, \vartheta) = \begin{cases} 0, & \text{if } \xi \leq \ell(\mathfrak{D}_{\hat{\tau}_1}^T, \mathfrak{D}_{\hat{\tau}_1}^C; \vartheta) \\ \frac{k}{\mathcal{B}}, & \text{if } \ell(\mathfrak{D}_{\hat{\tau}_k}^T, \mathfrak{D}_{\hat{\tau}_k}^C; \vartheta) < \xi \leq \ell(\mathfrak{D}_{\hat{\tau}_{k+1}}^T, \mathfrak{D}_{\hat{\tau}_{k+1}}^C; \vartheta) \quad (k = 1, 2, \ldots, \mathcal{B}) \\ 1, & \text{if } \ell(\mathfrak{D}_{\hat{\tau}_\mathcal{B}}^T, \mathfrak{D}_{\hat{\tau}_\mathcal{B}}^C; \vartheta) < \xi. \end{cases} \tag{35}$$

Based on Proposition (2) and $\kappa_\alpha = \max\{\frac{2-\alpha}{1-\alpha}, \frac{\alpha}{1-\alpha}\}$, we know the inequality holds:

$$\varphi_\alpha(\hat{\xi}_\alpha(\vartheta); \vartheta) - \mathcal{E}_\alpha(\vartheta) < \kappa_\alpha \delta. \tag{36}$$

With Eq. (34)/(36), the following inequality naturally holds when $\mathcal{B}$ is large enough:

$$\varphi_\alpha(\hat{\xi}_\alpha; \vartheta_*) \leq \mathcal{E}_\alpha(\vartheta_*) + \kappa_\alpha \left[ \frac{\alpha - \hat{F}_\ell(\xi_\alpha; \mathcal{B}, \vartheta_*)}{\frac{dF_\ell(\xi;\vartheta)}{d\xi}|_{\xi=\xi_\alpha}} + \mathcal{R}_\mathcal{B} \right].$$

# J  Experimental Set-up & Implementation Details

This section is to provide experimental details in this paper. For the implementation of few-shot sinusoid regression and few-shot image classification, we respectively refer the reader to TR-MAML's codes (https://github.com/lgcollins/tr-maml) in (Collins et al., 2020) and vanilla MAML's codes (https://github.com/AntreasAntoniou/HowToTrainYourMAMLPytorch) in (Antoniou et al., 2019). And ours is built on top of the above codes except for simple modification of loss functions. The learning rates for the inner loop and the outer loop of all methods are the same as the above ones.

To facilitate the use of our heuristic optimization strategy, we leave the pytorch version of loss functions within the expected tail risk minimization. The example is provided in the case of mean square errors after MAML's inner loop, which is *simple to implement yet effective in robustifying fast adaptation*, as follows:

```
1  import torch
2  from torch.nn import MSELoss
3
4  def cvar_mse(y_pred,y_true,conf_level=0.5):
5
6      batch_MSE=MSELoss(reduction='none')
7      batch_loss=batch_MSE(y_pred,y_true)
8
9      # average risk values over non-task dimensions
10     batch_avg_loss=torch.mean(batch_loss,dim=-1)
11
12     # crude Monte Carlo to estimate VaR and sub-tasks
13     topk_mse,topk_idxs=torch.topk(batch_avg_loss,int((1-conf_level)*
       y_true.size()[0]))
14
15     return torch.mean(topk_mse)
```

Listing 1: Loss Functions in Two-Stage Heuristic Algorithm with Crude Monte Carlo

## J.1 Meta Learning Datasets & Tasks

**Sinusoid Regression.** MAML (Finn et al., 2017), TR-MAML (Collins et al., 2020), and DR-MAML are considered in this experiment. We retain the setup in task generation and partition in (Collins et al., 2020). More specifically, there exists a distribution drift between the meta-training and the meta-testing function families $\{f_m(x) = a_m \sin(x - b_m)\}_{m=1}^M$.

Numerous easy tasks and a small proportion of difficult tasks are available in meta-training, while all tasks in the space are used in the evaluation. The range of the phase parameter is $b \in [0, \pi]$, and the amplitude range of the parameter is $a \in [0.1, 1.05]$ for easy tasks and $a \in [4.95, 5.0]$ for difficult tasks. It is noted that the sinusoid task is more challenging to fit with larger amplitudes as the resulting function is less smooth. The loss function corresponds to the mean-squared error between the predicted value $f(x)$ and the ground truth value. The number of task batches is 50 for 5-shot and 25 for 10-shot. The optimal selection of the confidence level $\alpha$ is difficult since we need to trade off the worst and average performance. Our setup is to minimize $\text{CVaR}_\alpha$, which already considers the worst-case at some degree, so we watch the average performance in meta training results and set $\alpha = 0.7$ for all few-shot regression tasks. There is no external configuration for this hyper-parameter. The maximum number of iterations in meta-training is 70000.

**Few-shot Image Classification.** MAML (Finn et al., 2017), TR-MAML (Collins et al., 2020), and DR-MAML are considered in this experiment. The `N-way K-shot` classification corresponds to an `N`-classification problem with `K`-labeled examples available to the meta learner.

The Omniglot dataset consists of 1623 handwritten characters from 50 alphabets, with each 20 examples. The task distribution is uniform for all task instances consisting of characters from one specific alphabet. The dataset split follows procedures in (Triantafillou et al., 2019). Finally, 25 alphabets are used for meta-training, with 20 alphabets for meta-testing. The number of task batches is 16. The confidence level $\alpha = 0.5$ is selected with the same criteria as that in few-shot regression tasks. The maximum number of iterations is 60000 in meta-training. As the construction of the Omniglot meta dataset is related to specific alphabets and the scale of combination for tasks is huge, this indicates randomly sampled meta-training tasks in the evaluation of the main paper Tables may not be used in meta-training.

The *mini*-ImageNet dataset is pre-processed according to (Larochelle, 2017). In detail, 64 classes are used for meta-training, with the remaining 36 classes for meta-testing. Tasks are generated as follows: 64 meta-training classes are randomly grouped into 8 meta-train tasks with the class numbers $\{6, 7, 7, 8, 8, 9, 9, 10\}$, and the 36 meta-testing classes are processed in the same way. Finally, each task is built by sampling 1 image from 5 different classes within one task, resulting in a `5-way 1-shot` problem. The number of task batches is 4. The maximum number of iterations is 60000 in meta-training.

## J.2  Neural Architectures & Optimization

**Sinusoid Regression.** MAML (Finn et al., 2017), TR-MAML (Collins et al., 2020), and DR-MAML are considered in this experiment. We retain the neural architecture for regression problems in (Finn et al., 2017; Collins et al., 2020). That is, we deploy a fully-connected neural network with two hidden layers of 40 ReLU nonlinear activation units. All methods use one stochastic gradient descent step as the inner loop.

**Few-shot Image Classification.** We retain the neural architecture for few-shot image classification problems in (Finn et al., 2017; Collins et al., 2020). In detail, a four-layer convolutional neural network is used for both Omniglot and *mini*-ImageNet datasets. All methods use one stochastic gradient descent step as the inner loop.

# K  Additional Experimental Results

**More Quantitative Analysis.** Due to the page limit in the main paper, we include the $\alpha$'s sensitivity experimental result in sinusoid `10-shot` regression. As illustrated in Fig. (7), the trend is similar to that in sinusoid `5-shot` regression. Worst-case optimization degrades the average performance of TR-MAML. DR-MAML is entangled with MAML in the average performance, while the performance gap between them is significant in the worst-case.

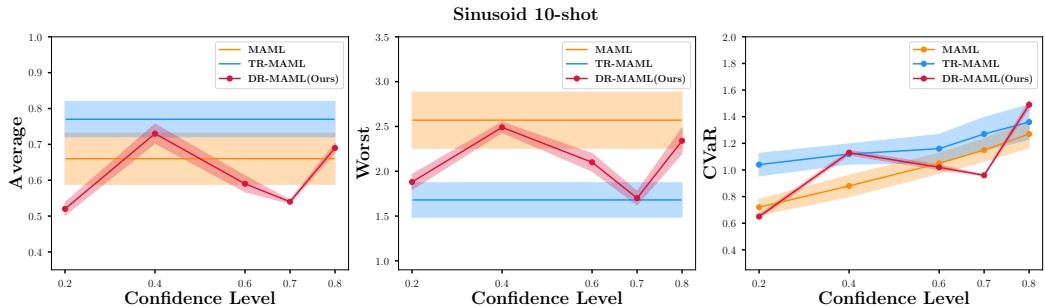

Figure 7: **Meta Testing Performance of Meta-Trained DR-MAML with Various Confidence Levels $\alpha$.** MAML and TR-MAML are irrelevant with the variation of $\alpha$ in meta-training. The plots report testing MSEs with standard error bars in shadow regions.

Regarding few-shot image classification in the *mini*-ImageNet dataset, we can observe that in Fig. (8), the standard error is relatively smaller than in previous regression cases. When the confidence level is over a particular value, *e.g.* $\alpha > 0.5$, there occurs a significant decline of performance in all metrics. Note that when $\alpha \to 1.0$, the optimization objective approaches the worst-case optimization objective. Here we attach two possible reasons for the performance degradation phenomenon: (i) The adopted base optimization technique matters in nearly worst-case optimization. The stochastic mirror descent-ascent (Juditsky et al., 2011) is utilized in TR-MAML, which is more stable in deriving the optimal solution. In comparison, the stochastic gradient descent with sub-gradient operations works as the optimization method, and this method can be unstable when the scale of worst-case examples is small in the update. (ii) For few-shot image classification, estimates of $\text{VaR}_\alpha$ can be less precise with limited batch sizes and higher $\alpha$ values since the meta risk function value is discontinuous. Consequently, we can also attribute the severe degradation of fast adaptation performance in higher $\alpha$ value cases to the approximation errors of quantile estimates.

**More Visualization Results.** Further, we explore the influence of the expected tail risk minimization principle in meta learning. Here the landscape of meta risk values, namely fast adaptation losses, is presented in the sinusoid regression problem.

As exhibited in Fig. (9), the evaluated meta risk values from one random trial are associated with hyper-parameters of tasks. The final optimized results can discover some tasks difficult in fast adaptations. Meta learning methods are difficult to adapt in task regions with higher amplitudes. MAML exhibits higher MSEs in regions with the amplitude $a > 2$, while DR-MAML minimizes a proportion of risks in these regions. In contrast, TR-MAML reduces the risk around task regions with $a > 2$ to a certain extent; however, it shows relatively higher risk values in easy regions. Such

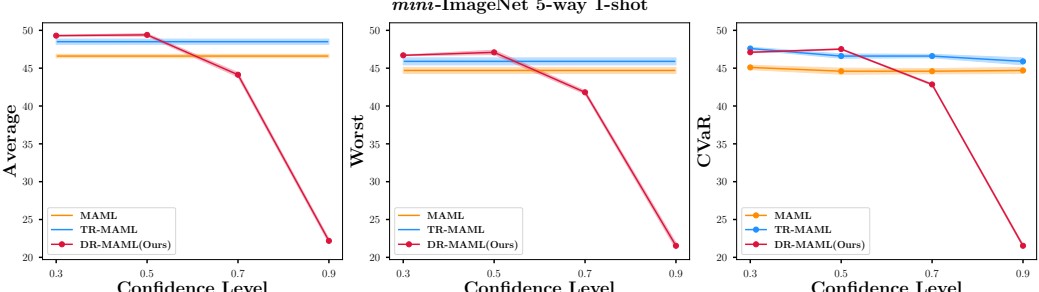

Figure 8: **Meta Testing Classification Accuracies of Meta-Trained DR-MAML with Various Confidence Levels** $\alpha$. MAML and TR-MAML are irrelevant with the variation of $\alpha$ in meta-training. The plots report testing accuracies with standard error bars in shadow regions.

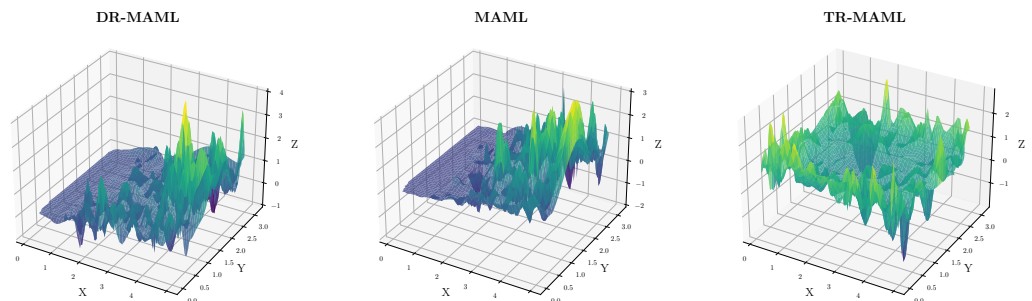

Figure 9: **The Fast Adaptation Risk Landscape of Meta-Trained DR-MAML, TR-MAML and MAML.** Shown is an example of sinusoid 5-shot regression, which corresponds to the function space $f(x) = a\sin(x - b)$. The $X$-axis denotes the amplitude parameter $a$, and the $Y$-axis is the phase parameter $b$. The confidence level is $\alpha = 0.7$ in meta-training. The plots report testing MSEs in the $Z$-axis with a random trial of generating tasks.

evidence reflects the interpretability in optimization within the expected tail risk minimization, and the landscape of meta risk values is relatively flat and smooth than others.

**Experimental Results with DR-CNPs.** The implementation is the same as CNP (Garnelo et al., 2018a) and official Github files (`https://github.com/deepmind/neural-processes/blob/master/conditional_neural_process.ipynb`); the Gaussian Process curve generator works as the benchmark, and we retain the neural architecture and optimization pipelines in Github files. Please check implementation details, *e.g.*, neural architectures, optimizers, epochs, batch sizes, *etc.*, from the mentioned GitHub repository.

Also, note that this is more challenging than sinusoid as there is more randomness in generating curves. Similar to few shot image classifications, we set $\alpha = 0.5$ in meta training DR-CNPs. In meta-testing processes, we randomly sample 64 curves per run with random context points in GitHub files and examine the performance of CNP, TR-CNP (worst-case), and DR-CNP (distributionally robust). The log-likelihoods in 5 runs are reported as follows (the higher, the better).

From Table (6), we observe that the DR-CNP retains the average performance the same as the CNP and simultaneously achieves the highest worst and $\text{CVaR}_\alpha$ log-likelihoods. In contrast, the TR-CNP sacrifices the average performance a lot and obtains the mediate worst result. Here we attribute

Table 6: **Meta-testing point-wise log-likelihood results of Gaussian processes in 5 runs.**

| Method | Average | Worst | CVaR$_\alpha$ |
|---|---|---|---|
| CNP (Garnelo et al., 2018a) | **-0.22**$_{\pm \textbf{0.04}}$ | -1.35$_{\pm 0.38}$ | -0.50$_{\pm 0.11}$ |
| TR-CNP | -0.80$_{\pm 0.01}$ | -1.08$_{\pm 0.02}$ | -0.87$_{\pm 0.01}$ |
| DR-MAML | **-0.21**$_{\pm \textbf{0.02}}$ | **-0.74**$_{\pm \textbf{0.08}}$ | **-0.37**$_{\pm \textbf{0.03}}$ |

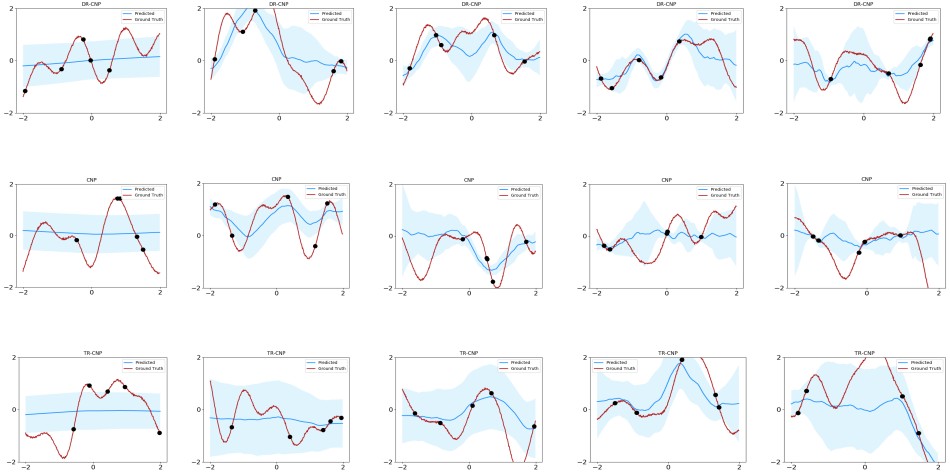

Figure 10: **Curve Fitting Result Visualization in 5 Random Trials.** From Up to Down are respectively DR-CNPs (Ours), CNPs, and TR-CNPs. Shadow regions are three standard deviations.

the failure of achieving the best worst performance using TR-CNPs to the sensitivity of min-max optimizers as mentioned in TR-MAML.

Particularly, some randomly sampled fitted curves in difficult scenarios, *e.g.*, setting the number of context points as 5, are visualized in Fig. (10). Observations are consistent with Table (6), where TR-CNPs exhibit over estimated uncertainty and cannot well reveal the trend of curves. In comparison, DR-CNPs can well handle challenging cases, capturing more convincing uncertainty.

**Experimental Results in Meta Reinforcement Learning.** We also performed additional experiments in 2-D point robot navigation tasks. This is a meta reinforcement learning benchmark. All setups of the point robot are the same as MAML, and please refer to pytorch-maml-rl and cavia Github for details of the navigation environments. The neural network of policies is a two-hidden layer MLP with 64 hidden units and tanh activations. The fast learning rate is 0.1, and the number of batches is 500. Other details, *e.g.*, optimizers, step-wise rewards, horizons and *etc.*, can be found in MAML. The one-step gradient is performed for fast adaptation, and the trust region policy optimization works as the policy gradient algorithm. Similar to few shot image classifications, we set $\alpha = 0.5$ in meta training point robots. In meta-testing processes, we randomly sample 100 navigation goals as tasks and examine the performance of MAML, TR-MAML, and DR-MAML. The histogram comparison is visualized in Fig. (11).

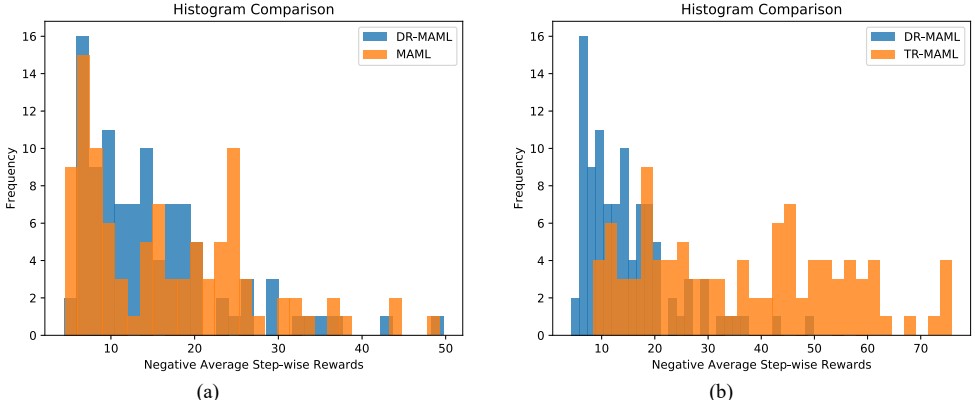

Figure 11: **Histograms of Meta-Testing Performance in Point Robot Navigation Problems.** We report the negative average step-wise rewards in $x$-axis.

Unlike few-shot supervised learning, worst-case optimization is pretty challenging in RL, and the result of TR-MAML is unstable and cannot achieve the goal of worst-case optimality in Table (7).

Table 7: **Meta-testing step-wise rewards in 4 runs in point robot navigation tasks.**

| Method | Average | Worst | $\text{CVaR}_\alpha$ |
|---|---|---|---|
| CNP (Garnelo et al., 2018a) | $-16.6_{\pm 0.19}$ | $\mathbf{-46.9_{\pm 4.22}}$ | $-24.2_{\pm 0.49}$ |
| TR-CNP | $-36.8_{\pm 0.12}$ | $-81.3_{\pm 4.35}$ | $-52.1_{\pm 0.89}$ |
| DR-MAML | $\mathbf{-15.3_{\pm 0.49}}$ | $\mathbf{-46.7_{\pm 3.24}}$ | $\mathbf{-21.6_{\pm 0.79}}$ |

This is due to the nature of large performance deviations when deploying policies in various tasks. Seeking optimizers for stable worst-case optimization is non-trivial. Also, stably generalization across skills in reinforcement learning is indeed difficult with a few interactions. Overall, DR-MAML can well control the proportional worst case performance.

## L   Platforms & Computational Tools

In this research project, we use NVIDIA 1080-Ti GPUs in computation. Pytorch (Paszke et al., 2019) works as the deep learning toolkit in implementing few-shot image classification experiments. Meanwhile, Tensorflow is the deep learning toolkit for implementing sinusoid few-shot regression experiments.

## M   Declaration of Author Contribution

The authors confirm their contribution to the paper as follows: Q.W. (Qi Wang) and Y.Q.L. (Yiqin Lv) conceptualized the idea of tail risk minimization in meta learning and developed simple yet effective strategies with mathematical demonstrations. Y.Q.L. performed most of the experiments and Q.W. examined the performance in meta reinforcement learning benchmark. Q.W. and Y.Q.L. wrote and prepared the manuscript, while Y.H.F. (Yanghe Feng) contributed crucial insights into the organization and checked the mathematical part of the draft in submission. J.C.H. (Jincai Huang) and Z.X. (Zheng Xie) contributed to the supervision and helped proofread the manuscript. All authors reviewed the results and approved the final version of the manuscript. Meanwhile, we also thank dr. Zhengge Zhou for helpful discussion.

