strategies consistently outperform that with the risk-weighted ones in both `5-shot` and `10-shot` sinusoid cases regarding all metrics. The performance advantage of using the two-stage ones is not significant in *mini*-ImageNet scenarios. We can hypothesize that the estimate of $\text{VaR}_\alpha$ in continuous task domains, *e.g.*, sinusoid regression, is more accurate, and this probabilistically ensures the improvement guarantee with two-stage strategies. Both the $\text{VaR}_\alpha$ estimate in two-stage strategies and the importance weight estimate in the risk-reweighted ones may have a lot of biases in few-shot image classification, which lead to comparable performance.

# L    Platforms & Computational Tools

In this research project, we use NVIDIA 1080-Ti GPUs in computation. Pytorch (Paszke et al., 2019) works as the deep learning toolkit in implementing few-shot image classification experiments. Meanwhile, Tensorflow is the deep learning toolkit for implementing sinusoid few-shot regression experiments.