# OpenReview forum: "A Simple Yet Effective Strategy to Robustify the Meta Learning Paradigm"
_NeurIPS.cc/2023/Conference — NeurIPS 2023 poster_

### Official Review · Reviewer_Wn6T · 2023-06-28

**Soundness:** 3 good
**Presentation:** 3 good
**Contribution:** 2 fair
**Rating:** 5
**Confidence:** 4

**Summary:**

The authors address an existing gap in fast adaptation of meta-learning by developing a robustification framework that draws from the concept of expected tail risk and recasts robustification as a distributional optimization problem. They adopt a heuristic two-stage optimization strategy that is broadly useful toward most meta-learning methods. The applicability of this framework toward enhancing robustness against task distributions and mitigating worst-case performance is verified and discussed over a selection of benchmark tasks.

**Strengths:**

+ The paper motivates the importance of expanding work in distributionally robust optimization and presents a unique formulation of the robustness problem for meta learning.
+ Concrete illustration of how the proposed optimization method can benefit existing meta learning work.
+ Well-organized sections and visual structure.
+ Nice FAQ — the authors should consider making their contributions clearer in the main text of the paper.

**Weaknesses:**

- Appendix J.1 states: "The optimal selection of the confidence level α is difficult since we need to trade off the worst and average performance. Our setup is to minimize CVaRα, which already considers the worst-case at some degree, so we watch the average performance in meta training results and set α = 0.7 for all few-shot regression tasks." This isn't really tail risk (e.g., alpha > 0.9, alpha < 0.05). Given the tail risk problem, why does alpha range from 0.2 to 0.8 (Figure 5)?

- The paper seems incremental to TR-MAML. It would have been better to see more detail in the main text (as opposed to Appendices) on the main contribution of taking a probabilistic approach to meta learning and using a cVaR probabilistic measure (e.g., applicability, convergence, bounds). Instead, the main text focused on implementing DR-MAML which may not work for true tail scenarios.

- The authors should introduce at least a new dataset versus just reusing MAML/TR-MAML. Appendix J shows that the authors simply modified the loss function, reusing code from MAML and TR-MAML. It would be nice to see additional experiments in more challenging settings, e.g., RL tasks like point navigation or continual learning.

- Appendix K states: "Note that when α → 1.0, the optimization objective approaches the worst-case optimization objective. Here we attach two possible reasons for the performance degradation phenomenon:..." This should be expanded and included in the main text as opposed to the Appendix.

- It seems the main contribution is imposing a probabilistic constraint (cVaR) versus using worst-case TR-MAML, which necessarily only works for meta-learning over a large task space.


**Questions:**

- What is the justification for deriving results over only 3 runs? This makes empirical claims seem suspect, e.g. the conclusion that “DR-MAML surpasses MAML in terms of average performance”

- It would be nice to see additional discussion of the impacts of assumptions 1/2/3. (The authors briefly touch on this around line 294.)

- Pseudocode is in Appendix B but it would be more useful in the main text.

* Experimental results for the proposed approach could be further substantiated by including a discussion of Tables 4/5 in the main text.

* Minor:
	* line 34: “plaint” -> “plain” (note: this may be ignorance on my part.)
	* line 199: “sometime” -> “sometimes”
	* line 208: “a” -> “an”
	* Figure 4 caption line 2: “DR-MAMAL” -> “DR-MAML”
	* Appendix, line 859: “Risk-Rweighted” -> “Risk-Reweighted”


**Limitations:**

* The authors name the primary theoretical and practical limitations of their work.

* Societal impacts are not directly applicable. The authors touch upon limitations of existing meta-learning methods for high-risk real-world scenarios, which is important.

---

> ### Author Rebuttal · Authors · 2023-08-08
>
> We sincerely thank \# Reviewer Wn6T for these insightful comments. The remainders focus on concerns to address.
>
> ***
> **1. Misconception of the tail risk**
>
> Thanks for the comment.
>
> (1) In our paper, ***the concept of the tail risk is from the literature work [1-2] and the standard definition in Wikipedia term “tail risk”***. The varying $\alpha$ describes the tail risk level. As for $\alpha$ ranging between 0.2 and 0.8, it is because we can only test limited scenarios in $\alpha\in(0,1)$, and $\alpha\to 0$ and $\alpha\to 1.0$ corresponds to the average and worst case.
>
> (2) Due to numerous tasks, the optimization strategies, including ours and group DRO, perform batch optimization in practice.
>
> Hence, the concept of the tail risk in this paper is correct and optimization in a task batch way works in the tail cases.
>
> **Reference**
>
> [1] Sarykalin, S., Serraino, G., & Uryasev, S. (2008). Value-at-risk vs. conditional value-at-risk in risk management and optimization. In State-of-the-art decision-making tools in the information-intensive age (pp. 270-294). Informs Tutorials in Operations Research.
>
> [2] Hong, L. J., & Liu, G. (2009). Simulating sensitivities of conditional value at risk. Management Science, 55(2), 281-293.
>
> ***
> **2. Incremental to TR-MAML**
>
> Thanks for the question. We need to clarify that our work differs a lot from TR-MAML:
>
> (1) define distributionally robustness for fast adaptation rather than worst-case in TR-MAML; (2) design heuristic two-stage optimization algorithms with improvement guarantee; (3) show increased robustness without sacrificing the average performance in contrast to TR-MAML.
>
> Hence, the method is new with theoretical contributions and balancing performance. Though the experimental set-up is incremental w.r.t. TR-MAML, we also provided new experimental results after taking advice.
>
> ***
> **3. Include the point navigation**
>
> Thanks for these constructive comments. We’ve taken advice and run experiments in point robots for navigation tasks, and will include results in the final version.
>
> (1) The 2-D point robot navigation is a meta reinforcement learning benchmark. All setups of the point robot are the same as MAML [1-2], and please refer to pytorch-maml-rl and cavia Github for environment and task details. The policy network is a two-hidden layer MLP with 64 hidden units and tanh activations. Other details, e.g., optimizers, step-wise rewards, horizons and etc., can be found in MAML [1]. The one-step policy gradient is performed for fast adaptation with the learning rate 0.1, and TRPO [3] works as the policy gradient algorithm. $\alpha=0.5$ is set in meta training DR-MAML. In meta-testing, we uniformly sample 100 navigation tasks (goals’ locations follows the distribution $U[-0.5, 0.5]^2$) and examine the performance. The step-wise episodic rewards are reported as follows.
>
> New results as Table 7. Meta-testing step-wise rewards in 4 runs in point robot navigation tasks (the higher, the better).
>
> |  Methods     | Average | Worst | $\text{CVaR}_{\alpha}$ |
> |--------------|---------|-------|------------------------|
> | MAML          | $-16.6(\pm0.19)$     | $-46.9(\pm4.22)$   | $-24.2(\pm0.49)$                    |
> | TR-MAML       | $-36.8(\pm0.12)$    | $-81.3(\pm4.35)$   | $-52.1(\pm0.89)$                    |
> | DR-MAML(Ours) | $-15.3(\pm0.49)$     | $-46.7(\pm3.24)$   | $-21.6(\pm0.79)$                    |
>
> Unlike few-shot supervised learning, worst-case optimization is pretty challenging in RL, and the result of TR-MAML is unstable and cannot achieve the goal of worst-case optimality in Table 7. This is due to the nature of large performance deviations when deploying policies in various tasks. Seeking optimizers for stable worst-case optimization is non-trivial.  Also, stably generalization across skills in reinforcement learning is indeed difficult with a few interactions. Overall, DR-MAML can well control the proportional worst case performance.
>
> ***
> **4. Whether method works for meta learning over a large task space**
>
> Thanks for the question. You are right, and we consider the distribution of tasks. The combinatorial complexity of tasks is high, meaning a large task space in our experiments.
>
> ***
> **5. Moving appendix contents to the main paper**
>
> Thanks for the comment. We’ll move the appendix pseudo code, the appendix K “Note that when $\alpha$ …”, discussions on Table 4/5 to the main paper.
>
> ***
> **6. Other questions**
>
> (1) Regarding impacts of assumptions 1/2/3, these support the derivation of remarks, propositions, and theorems. Their roles are reflected in Appendix E-H.
>
> (2) Regarding 3 runs in image classification experiments, there are two reasons: (a) for fair comparison, we keep all the experimental setups the same as TR-MAML. (b) another comes from the relatively stable classification performance in different time-consuming runs.
>
> ***
> **7. Typos**
>
> Thanks for the comment. We revised the “plaint” to “plain” in Line34; “sometime” to sometimes” in Line 199; a to an in Line 208, Figure4 caption “DR-MAMAL” to “DR-MAML” in Figure4 caption Line 2; and Appendix “Risk-Rweighted” to “Risk Reweighted” in Line 859.
>
> **Reference**
>
> [1] Finn, C., Abbeel, P., & Levine, S. (2017, July). Model-agnostic meta-learning for fast adaptation of deep networks. ICML.
>
> [2] Zintgraf, L., Shiarli, K., Kurin, V., Hofmann, K., & Whiteson, S. (2019, May). Fast context adaptation via meta-learning. ICML.
>
> [3] Schulman, J., Levine, S., Abbeel, P., Jordan, M., & Moritz, P. (2015, June). Trust region policy optimization. ICML.
>
> ***
> **8. Particular thanks**
>
> We express particular gratitude to \# Reviewer Wn6T for carefully proofreading throughout the main and the appendix. This indeed means a lot of effort and helps improve the quality of the manuscript. We hope your concerns are well addressed.

---

> > ### Comment · Reviewer_Wn6T · 2023-08-11
> > **Thanks**
> >
> > Thanks very much for the clarification and responses, and apologies for the misconception of CVaR on my part. The edits and new additions introduced are helpful to boost the paper, and my concerns have been addressed. I will raise my score accordingly.

---

> > > ### Author Response · Authors · 2023-08-11
> > >
> > > Once again, we thank \# Wn6T for the positive comments and updates of reviews.

---

### Official Review · Reviewer_vUMr · 2023-06-29

**Soundness:** 2 fair
**Presentation:** 3 good
**Contribution:** 3 good
**Rating:** 6
**Confidence:** 4

**Summary:**

The paper proposes a novel expected tail risk objective for meta-learning, which provides an intermediate of the worst-case and average risk minimization algorithms. The proposed algorithm can be adapted to existing meta-learning algorithms with improved robustness.

**Strengths:**

* The paper is well-written and easy to follow.
* Considering expected tail risk in meta-learning is novel to me. Controlling the worst task performance at a certain probabilistic level can improve the robustness.
* The experimental results show the effectiveness of the proposed approach.


**Weaknesses:**

* The meta risk function $\ell(D_{\tau}^T, D_{\tau}^C;\nu)$ is an empirical risk estimation that depends on the number of data points $m$ of task $\tau$. It’s preferable to consider its generalization as in [1,2,3].
* Why only adopt eight meta-training tasks and four meta-testing tasks for the experiment of mini-ImageNet in Table.3?

>[1] Giulia Denevi, Carlo Ciliberto, Riccardo Grazzi, and Massimiliano Pontil. Learning-to-learn stochastic gradient descent with biased regularization. In International Conference on Machine Learning, pages 1566–1575. PMLR, 2019a.

>[2] Qi Chen, Changjian Shui, and Mario Marchand. Generalization bounds for meta-learning: An information-theoretic analysis. Advances in Neural Information Processing Systems, 34: 25878–25890, 2021.

>[3] Yu Bai, Minshuo Chen, Pan Zhou, Tuo Zhao, Jason Lee, Sham Kakade, Huan Wang, and Caiming Xiong. How important is the train-validation split in meta-learning? In International
Conference on Machine Learning, pages 543–553. PMLR, 2021.

**Questions:**

See the discussion in the weaknesses section.

**Limitations:**

Yes.

---

> ### Author Rebuttal · Authors · 2023-08-06
>
> We sincerely thank \# Reviewer vUMr for these constructive comments. The remainders focus on questions to answer.
>
> ***
> **1. Consider and discuss generalization related work.**
>
> Thanks for this suggestion.
>
> (1) Most previous work studies the generalization in the context of the complete task space within the expected risk minimization principle, while our work considers the expected tail risk. This set-up difference brings more difficulties in accurately quantifying the bound. *Instead, this work pays more attention to the estimated stepwise bound w.r.t. $\alpha$ in Proposition 2, the improvement guarantee and convergence properties in Theorem1, and the gaps of optimized solution in Theorem2. As for generalization capability across tasks, we execute the quantitatively evaluation in experiments.*
>
> (2) In Literature Review section Line83, *we will cite and add more discussions on references [1-3] in the final version* as follows:
>
> Insert the contents in Line 84
>
> >Besides, there exist several important works investigating the generalization capability of methods. \citet{bai2021important} [3] conduct the theoretical analysis of the train-validation split and connect it to optimality. In \citep{chen2021generalization} [2], a generalization bound is constructed for MAML through the lens of information theory. \citet{denevi2019learning} [1] study an average risk bound and estimate the bias for improving stochastic gradient optimization in meta learning.
>
> **Reference:**
>
> [1] Giulia Denevi, Carlo Ciliberto, Riccardo Grazzi, and Massimiliano Pontil. Learning-to-learn stochastic gradientdescent with biased regularization. In International Conference on Machine Learning, pages 1566–1575. PMLR,2019a.
>
> [2] Qi Chen, Changjian Shui, and Mario Marchand. Generalization bounds for meta-learning: An information-theoretic analysis. Advances in Neural Information Processing Systems, 34: 25878–25890, 2021.
>
> [3] Yu Bai, Minshuo Chen, Pan Zhou, Tuo Zhao, Jason Lee, Sham Kakade, Huan Wang, and Caiming Xiong. Howimportant is the train-validation split in meta-learning? In International Conference on Machine Learning, pages543–553. PMLR, 2021.
>
> ***
> **2. Setup of tasks in mini-imageNet**
>
> Thanks for this question.
>
> (1) In the original TR-MAML’s mini-imageNet setup, the concept of eight meta-training and four meta-testing is in the combinatorial class number sense. For example, since there 100 classes, the eight meta-training means the partition of 64 classes with task lists $\{6,7,7,8,8,9,9,10\}$, where the element corresponds to the number of classes in one task. Other M-way N-shot setting is standard in the domain.
>
> (2) For a fair comparison, we use the same setup as TR-MAML, where eight meta-training and four meta-testing tasks are created for mini-imageNet experiments.
>
> ***
> Finally, we hope your questions are well answered. And many thanks for your helpful reviews. For any other question, we are happy to discuss it and make further clarifications.

---

> > ### Comment · Reviewer_vUMr · 2023-08-16
> >
> > Thanks for your clarifications and responses. Despite the lack of a more rigorous generalization analysis, considering the expected tail risk to robustify the meta-learning process is beneficial.
> >
> > So I tend to keep the current score and recommend an acceptance.

---

> > > ### Author Response · Authors · 2023-08-16
> > >
> > > Thank you for your helpful suggestions and kindness. Your comments help improve the manuscript a lot.

---

### Official Review · Reviewer_NA6N · 2023-07-03

**Soundness:** 2 fair
**Presentation:** 3 good
**Contribution:** 2 fair
**Rating:** 5
**Confidence:** 4

**Summary:**

This paper considered a robust meta-learning framework by addressing the worst-tasks performance. Specifically, this paper adopted conditional value at risk (CVR) to filter poorly performed tasks. In order to effectively learn the CVR, this paper adopted a surrogate loss, a variant of hinge loss with a learnable threshold. The proposed framework exhibits empirical improvement in certain few-shot tasks.


**Strengths:**

In general, this paper tackles an interesting problem in mete-learning. I would think the following points might be interesting.
- Addressing robustness issue in meta-learning.
- A principled and simple method through the tool of conditional value at risk.
- The paper is quite clearly written.


**Weaknesses:**

This reviewer found there are several concerns in the current version. The factors that make this reviewer suggest that a major revision is better. I will discuss these points in the following aspects.


**Originality** and **Significance**

The main motivation of the proposed method is to address the robustness in meta-learning. However, this reviewer feels a bit confused about its motivation in choosing CVR based approaches. There are alternative methods in distribution robust optimization such as group dro, which can be naturally extended by adding the task weights as a learnable parameter. Or if we adopt Wasserstein distance, we could generate new samples as a data augmentation training.

Based on these, I would think the proposed method may lack sufficient motivation or comparison to justify its benefits.


**Quality**


I have some technical concerns.

- In equation (2), the worst performing group sometimes is quite risky since it would have quite arbitrary value. It should be better to consider the robustness in the context of distributional robustness optimization.
- In sec 4.3, I would think there are better principles to learn the threshold value. I mean, since the learnable thresholding value is convex for the loss in (6), thus we could directly adopt several solvers to solve the optimal value after each epoch. This could construct a principled method to optimize the hyper-parameter.
- In the few-shot context, the worst-performing groups in general have quite few data, does the accuracy really make some statistical sense?
- The representation learning based meta-method seems not empirically validated?


**Clarity**.

Most parts within the paper are clear. Some parts are still a bit confusing, for example
- Sec 4.1 128-129 For any \theta, the meta learning.. It should be for any **fixed** \theta, right?
- The surrogate seems quite similar to the hinge loss, right? I would think a further discussion is better.
- DN-CRP is better named as meta-representation learning.
- Fig 2 seems a bit confusing because of \alpah_{t}, actually \alpha here is fixed. The threshold value was changing w.r.t. t rather than \alpha.







**Questions:**

In general I would think this paper has several merits. However, this reviewer has certain concerns on the novelties and technical parts. Thus currently I vote for a weak rejection. Authors are suggested to check weakness points to know details.

---

> ### Author Rebuttal · Authors · 2023-08-06
>
> We sincerely thank \# Reviewer NA6N for these insightful comments. The remainders focus on concerns to address.
>
> ***
> **1. Group DRO baseline comparison, research motivation and data augmentation**
>
> Thanks for this precious question.
>
> (1) Sorry for confusing you. ***The Group DRO baseline can be found in Appendix-K Line846-867, where it is called the risk reweighted method in Table 4/5***. The experimental analysis is attached there.
>
> (2) Regarding motivation, the choice of CVaR considers scenarios similar to group DRO. The difference is CVaR considers the tail risk while group DRO assigns the adjustable weight to tasks, and both are crucial for increasing robustness. Also, we build up connections between CVaR and group DRO in Appendix. Importantly, we contribute the improvement guarantee in a theoretical sense for analyzing CVaR objectives.
>
> (3) Since all baselines are not data augmentation-based methods, we only compare the risk minimization principles for fairness. We agree that data augmentation, like Wasserstein distance, has the potential for robustness improvement, and we leave it a future exploration.
> To avoid confusion, we will rename the risk reweighted by group DRO in the appendix.
>
> ***
> **2. Other principled method to optimize the hyper-parameter $\xi$**
>
> Thanks for this comment.
>
> (1) For the threshold value or slack variable $\xi$, we agree that other methods exist to optimize, such as deep density learning models. However, our proposed one, a quantile estimate with crude MC, is the simplest.
>
> (2) Note that the risk function is non-convex in most scenarios; our proposed heuristic method is easy to implement and has an improvement guarantee. Based on these properties and findings, we take the simple yet effective way for optimization.
>
> ***
> **3. Meta learning and statistical sense of few data**
>
> Thanks for the question. Meta learning or few-shot learning indeed relies on a few data points for skill transfer. As for the statistical sense, it is considered in a task distribution sense. Though data are few, the task space is huge, and statistical sense is verified in most meta learning work, including MAML.
>
> ***
> **4. Representation based meta learning to validate**
>
> Thanks for this suggestion. We’ve taken advice and included the DR-CNPs as the additional experiments. Please refer to **Response to \# Reviewer YmWX** or **Global Comments** for details.
>
> ***
> **5. Other clarity issues**
>
> Thanks for these points.
>
> (1) Regarding Line128-129, your understanding is correct, and it is fixed and bounded.
>
> (2) Regarding the hinge loss, our optimization objective indeed relates to that a bit. We'll add, “Note that our optimization objective also has a connection with hinge losses. Hinge losses are considered in a data point sense while the hinge operation in ours is examined in a risk value distribution sense.” in Line157.
>
> (3) Regarding DR-CNP, you are right, and it is a meta-representation learning method. We’ll further detail this in the final version.
>
> (4) Regarding $\alpha_t$, here $t$ is associated with $\xi$ and indexes the iteration number. Your suggestion is nice, and we will modify the location of indices in the Figure.
>
> ***
> Finally, we hope your questions are well answered. And many thanks for your helpful reviews. For any other question, we are happy to discuss it and make further clarifications.

---

> > ### Comment · Reviewer_NA6N · 2023-08-14
> >
> > Dear authors,
> >
> > Thanks for your rebuttal. I will spend a bit more time to read the rebuttal and a quick paper recap, since it has been one month after submitting the paper review. But I will ensure that your rebuttal will be definitely considered.. Thank you!

---

> > > ### Author Response · Authors · 2023-08-14
> > >
> > > Thanks for your time and kindness. Looking forward to your feedback.

---

> > > > ### Comment · Reviewer_NA6N · 2023-08-16
> > > >
> > > > Thank you! My concerns on group-DRO have been addressed and score has been updated.  I would suggest a discussion on DRO in the main paper will be beneficial.

---

> > > > > ### Author Response · Authors · 2023-08-17
> > > > > **Thanks for raising the scores**
> > > > >
> > > > > We would like to thank Reviewer NA6N for the positive feedback, and *we'll move the appendix discussion on group DRO to the main paper in the final version*.

---

### Official Review · Reviewer_ANpj · 2023-07-06

**Soundness:** 3 good
**Presentation:** 3 good
**Contribution:** 3 good
**Rating:** 7
**Confidence:** 3

**Summary:**

In this paper, the authors present a method by which to improve the expected tail risk during meta learning via a two-stage optimisation strategy. In addition, the authors provide theoretical and empirical analysis of their proposed approach, demonstrating improvements in the average, worst-case and conditional value at risk (CVar) metrics.

**Strengths:**

The paper is written and structured very well. In particular, Sections 2,3, and 4 provide great introductions to the meta-learning paradigm and the idea of risk-minimisation applied to meta learning. The theoretical analysis outlined in Section 4 is also quite clear. Looking at the supplementary material, proofs and assumptions are outlined well, and ablations are provided.

**Weaknesses:**

- This paper is quite strong, and many of my initial concerns are answered in the supplementary material.
- Besides minor issues with regards to wording and grammar, this paper is quite well written. For example, on lines 34-35, 'suffering from tribulation' and 'plaint'.
-  A minor concern, but one which might help practitioners. Choosing the optimal meta-batch size appears quite important to the final performance. Although the authors discuss the influence of task batch size, perhaps ablating across more (and a wider range of) batch sizes might be more informative.

For other weaknesses/comments, see questions.

**Questions:**

- In the paper, the Monte Carlo estimate allows for the theoretical analysis, but it is indeed quite crude. Do the authors have any comments on perhaps improving upon this (even if the theory isn't there yet)?
- What optimizers did the authors use? In the existing limitations section, they suggest that the batch size selection has a greater effect for first-order optimizers. Did the authors attempt to use any higher order (or approximate) optimizers?
- I believe the theoretical analysis breaks down for the image classification tasks? (please do correct me if I'm wrong) Do the authors have any further insights on this point, in particular as to why the tail risk is not well optimised for omniglot, but works well for mini-ImageNet?
- The authors also state that challenging meta-learning tasks can reveal more advantages of DR-MAML (line 281). Have more challenging tasks been tried? This could, for example be the meta-dataset or something equivalent?
- In Figure 5, there appears to be deviations from the trend when $\alpha = 0.7$, and in general, the trend appears to be somewhat non-monotonic. Is this a dataset dependant issue?
- Do the authors have any general guidelines for how the image classification performance of DR-MAML changes with $\alpha$?

**Limitations:**

The limitations of this work are outlined in Section 6 and assumptions are outlined throughout the paper. In addition, error bars and std's are provided for all experiments.

Overall, I see no potential negative societal impacts of this work. If anything, robust meta-learning has a potentially positive societal impact for deployed meta-learning systems.

---

> ### Author Rebuttal · Authors · 2023-08-08
>
> We sincerely thank \# Reviewer ANpj for these helpful comments. The remainders focus on questions to answer.
>
> ***
> **1. Potential improvement on crude Monte Carlo**
>
> This is a crucial question.
>
> (1) We choose the crude Monte Carlo in quantile estimate due to its simplicity. However, accurate estimation of the quantile contributes to better convergence in a theoretical sense. We can also use some deep density estimators for VaR estimates, which may require additional modeling and computations.
>
> (2) Another factor related to the crude Monte Carlo is the task batch, which is quite empirical. The main paper reports the sinusoid investigation using larger task batches greater than 50. For even larger batch sizes, especially in image cases, our device suffers from cuda out-of-memory issues.
>
> ***
> **2. More challenging tasks to investigate**
>
> (1) The 2-D point robot navigation is a meta reinforcement learning benchmark. All setups of the point robot are the same as MAML [1-2], and please refer to pytorch-maml-rl and cavia Github for environment and task details. The policy network is a two-hidden layer MLP with 64 hidden units and tanh activations. Other details, e.g., optimizers, step-wise rewards, horizons and etc., can be found in MAML [1]. The one-step policy gradient is performed for fast adaptation with the learning rate 0.1, and TRPO [3] works as the policy gradient algorithm. $\alpha=0.5$ is set in meta training DR-MAML. In meta-testing, we uniformly sample 100 navigation tasks (goals’ locations follows the distribution $U[-0.5, 0.5]^2$) and examine the performance. The step-wise episodic rewards are reported as follows.
>
> New results as Table 7. Meta-testing step-wise rewards in 4 runs in point robot navigation tasks (the higher, the better).
>
> |  Methods     | Average | Worst | $\text{CVaR}_{\alpha}$ |
> |--------------|---------|-------|------------------------|
> | MAML          | $-16.6(\pm0.19)$     | $-46.9(\pm4.22)$   | $-24.2(\pm0.49)$                    |
> | TR-MAML       | $-36.8(\pm0.12)$    | $-81.3(\pm4.35)$   | $-52.1(\pm0.89)$                    |
> | DR-MAML(Ours) | $-15.3(\pm0.49)$     | $-46.7(\pm3.24)$   | $-21.6(\pm0.79)$                    |
>
> Unlike few-shot supervised learning, worst-case optimization is pretty challenging in RL, and the result of TR-MAML is unstable and cannot achieve the goal of worst-case optimality in Table 7. This is due to the nature of large performance deviations when deploying policies in various tasks. Seeking optimizers for stable worst-case optimization is non-trivial.  Also, stably generalization across skills in reinforcement learning is indeed difficult with a few interactions. Overall, DR-MAML can well control the proportional worst case performance.
>
> (2) We’ve also performed additional experiments in stochastic process modelling with DR-CNP. Please refer to Response to \# Reviewer YmWX or Global Response for details.
> Contents to insert in Appendix K: Table 6. Meta-testing point-wise log-likelihood results in 5 runs (the higher, the better).
>
> |  Methods     | Average            | Worst              | $\text{CVaR}_{\alpha}$ |
> |--------------|--------------------|--------------------|------------------------|
> | CNP          | $-0.22(\pm0.04)$      | $-1.35(\pm0.38)$      | $-0.50(\pm0.11)$          |
> | TR-CNP       | $-0.80(\pm0.01)$      | $-1.08(\pm0.02)$      | $-0.87(\pm0.01)$          |
> | DR-CNP(Ours) | $-0.21(\pm0.02)$ | $-0.74(\pm0.08)$ | $-0.37(\pm0.03)$     |
>
> From Table 6, we observe that the DR-CNP retains the average performance the same as the CNP and simultaneously achieves the highest worst and $\text{CVaR}_{\alpha}$ log-likelihoods. In contrast, the TR-CNP sacrifices the average performance a lot and obtains the mediate worst result. Here we attribute the failure of achieving the best worst performance using TR-CNPs to the sensitivity of min-max optimizers as mentioned in TR-MAML.
>
> ***
> **3. Optimizers used in the experiments**
>
> (1) For a fair comparison, we keep all the experimental setups the same as TR-MAML. And first-order stochastic gradient descent is used in implementations for few-shot regression and classification scenarios.
>
> (2) As for higher order or more advanced optimizers, we run additional experiments in 2-D point robot navigation tasks, where the conjugate gradient method is used for trust region optimization. That is because standard first-order optimizers harm all baselines in convergence and stability.
>
> ***
> **4. Other empirical observations and guidelines**
>
> Thanks for your questions.
>
> (1) Regarding observed tail risk differences between Omniglot and mini-imageNet, we attribute this to a performance bottleneck. Note that Omniglot is relatively easy for all baselines. Hence all baselines achieve over 90% CVaR accuracies in meta-training. In contrast, DR-MAML can show more advantages in challenging tasks, e.g., mini-imageNet.
>
> (2) Regarding the deviations from the trend in Figure 5, we assume this is a data-dependent issue, and the quantile estimators may be less accurate in this scenario.
>
> (3) Regarding the general guidelines for $\alpha$ selection, in the image classification, we presume the mediate value, e.g., 0.5, is good enough. However, this also depends on the task batch size, as observed in the Appendix experiments.
>
> **Reference**
>
> [1] Finn, C., Abbeel, P., & Levine, S. (2017, July). Model-agnostic meta-learning for fast adaptation of deep networks. In International conference on machine learning (pp. 1126-1135). PMLR.
>
> [2] Zintgraf, L., Shiarli, K., Kurin, V., Hofmann, K., & Whiteson, S. (2019, May). Fast context adaptation via meta-learning. In International Conference on Machine Learning (pp. 7693-7702). PMLR.
>
> [3] Schulman, J., Levine, S., Abbeel, P., Jordan, M., & Moritz, P. (2015, June). Trust region policy optimization. In International conference on machine learning (pp. 1889-1897). PMLR.
>
> ***
> *Finally, we hope your questions are well answered. And many thanks for your reviews.*

---

> > ### Comment · Reviewer_ANpj · 2023-08-11
> > **Rebuttal Response**
> >
> > Dear Authors,
> >
> > Thank you for your detailed response.
> >
> > 1. Using a deep density estimation (something like: https://arxiv.org/abs/2107.11085) approach for VaR is interesting. I wonder if this might help or hinder the memory issues?
> > 2. Additional experiments are always welcome.
> > 3. With the previous and additional experiments, I wonder what performance benefits could be achieved with better (more stable, higher order etc) optimisers, rather than taking the ones used for TR-MAML?
> > 4. I agree that moving beyond Omniglot is a good idea. Usually when benchmark performance goes beyond 90%, they become far less illustrative of the differences between methods.
> >
> > These points are perhaps something for future work, but this discussion is interesting nonetheless. With the impact/scope of this paper, I believe my initial score is correct, especially given the new results. If the authors disagree or have any additional comments, please feel free to comment below.

---

> > > ### Author Response · Authors · 2023-08-11
> > >
> > > Thank you a lot for helpful feedback.
> > >
> > > We'll cite and discuss the deep density estimation (https://arxiv.org/abs/2107.11085)) in the final verision and the anwser to the memory issues requires future exploration. For higher order optimizers, stablizing the worstr or proportional worst cases can be interesting to investigate and we assume there are more advanced relaxation algorithms for solving this.
> > >
> > > Finally, your suggestions help improve our manuscript a lot. Many thanks.

---

### Official Review · Reviewer_YmWX · 2023-07-11

**Soundness:** 3 good
**Presentation:** 3 good
**Contribution:** 3 good
**Rating:** 6
**Confidence:** 3

**Summary:**

This paper proposes a novel meta-learning pipeline from the perspective of distributionally robust perspective to robustify fast adaptation. Through a two-stage strategy heuristically, the model could control the worst fast adaptation cases at a certain probabilistic level. And experiments show the effectiveness of the proposed model compared to other baselines, especially TR-MAML.

**Strengths:**

1. The proposed idea is interesting, and it could generalize the expected risk minimization and worst-case risk minimization for meta learning together.
2. The presentation and structure of the paper is good.
3. The experiments are extensive for the proposed model DR-MAML.

**Weaknesses:**

1. It is a difficult to follow for section 4. Some motivations or intuitions are necessary. For instance:
- what is the goal to introduce $(R^{+}, B)$ above line 129? It does not appear in the following paper.
- what is the intuition to involve the slack variable $\xi$ below line 157?

2. Some notations are not clear. For instance:
- line 134: why $M^{-1}(\ell)$ instead of $M(\ell)$?
- For Eq.(1) and (3), is it better to add "min" after the ":="?
- line 208: "a optimization..." -> "an opt..."

3. Do you consider diverse tasks during meta-training or just follow the common setting used in MAML? Since the goal is to robustify the fast adaptation, if you just use the common setting in MAML, how could you validate whether your model is helpful to keep robustness? Or the common setting is not robust?

4. Could you add some experiments for Example 2: DR-CNP?

**Questions:**

see weakness.

**Limitations:**

yes

---

> ### Author Rebuttal · Authors · 2023-08-08
>
> We sincerely thank \# Reviewer YmWX for insightful comments. The remainders focus on questions to answer.
> ***
> **1. Goal to introduce $(R^+,B)$ and the slack variable in Line157**
>
> Thanks for this question.
>
> (1) The goal of introducing $(R^+,B)$ is to describe the risk value distribution in the task space, and we need to induce the probability measure of risk values via the task distribution. This notation serves the CVaR definition.
>
> (2) The involvement of the slack variable is to relax the intractable unparameterized tail task distribution $p_{\alpha}(\tau;\vartheta)$ to the whole task distribution. Since the optimal of the slack variable is the VaR, this motivates the two-stage strategy as heuristics.
>
> ***
> **2. Notation meanings and typos**
>
> Thanks for these comments and suggestions.
>
> (1) In line134, the notation $\mathcal{M}^{-1}$ denotes the map from the risk value in the Euclidean space to the task space, as we need to screen the subspace of tasks in the tail risk for optimization.
>
> (2) We will modify “:=” after Equations in the final version and revise the typo “a optimization” to “an optimization” in line 208.
>
> ***
> **3. Diverse tasks or the common setting in meta-training**
>
> This is a crucial question in the literature. We use diverse tasks during meta-training, and this follows the same setup as TR-MAML. In TR-MAML, a large number of easy tasks and a few difficult tasks are used in meta-training. This setup in TR-MAML is better to validate the robustness as more difficult tasks appear in meta-testing processes. More explanations can be found in the setup section in TR-MAML.
>
> ***
> **4. Additional experiments for DR-CNP**
>
> Thanks for this nice suggestion. ***We’ve taken your advice and changed the manuscript by adding the DR-CNP results***. The implementation is the same as CNP [1] and official  deepmind/neural-processes Github files; the Gaussian Process curve generator works as the benchmark, and we retain the neural architecture and optimization pipelines in Github files. Please check implementation details, e.g., neural architectures, optimizers, epochs, batch sizes, etc., from the mentioned GitHub repository.
>
> Also, note that this is more challenging than sinusoid as there is more randomness in generating curves. Similar to few shot image classifications, we set $\alpha=0.5$ in meta training DR-CNPs. In meta-testing processes, we randomly sample 64 curves per run with random context points as generate_curves in GitHub files and examine the performance of CNP, TR-CNP (worst-case), and DR-CNP (distributionally robust). The log-likelihoods in 5 runs are reported as follows (the higher, the better):
>
> ***Contents to insert in Appendix K: Table 6. Meta-testing point-wise log-likelihood results in 5 runs.***
>
> |  Methods     | Average            | Worst              | $\text{CVaR}_{\alpha}$ |
> |--------------|--------------------|--------------------|------------------------|
> | CNP          | $-0.22(\pm0.04)$      | $-1.35(\pm0.38)$      | $-0.50(\pm0.11)$          |
> | TR-CNP       | $-0.80(\pm0.01)$      | $-1.08(\pm0.02)$      | $-0.87(\pm0.01)$          |
> | DR-CNP(Ours) | $-0.21(\pm0.02)$ | $-0.74(\pm0.08)$ | $-0.37(\pm0.03)$     |
>
> From Table 6, we observe that the DR-CNP retains the average performance the same as the CNP and simultaneously achieves the highest worst and $\text{CVaR}_{\alpha}$ log-likelihoods. In contrast, the TR-CNP sacrifices the average performance a lot and obtains the mediate worst result. Here we attribute the failure of achieving the best worst performance using TR-CNPs to the sensitivity of min-max optimizers as mentioned in TR-MAML.
>
> Particularly, ***some randomly sampled fitted curves in difficult scenarios, e.g., setting the number of context points as 5, are visualized in the Global Rebuttal PDF file, Figure 10***. Observations are consistent with Table 6, where TR-CNPs exhibit over estimated uncertainty and cannot well reveal the trend of curves. In comparison, DR-CNPs can well handle challenging cases, capturing more convincing uncertainty.
>
> ***
> **Reference**
>
> [1] Garnelo, M., Rosenbaum, D., Maddison, C., Ramalho, T., Saxton, D., Shanahan, M., ... & Eslami, S. A. (2018, July). Conditional neural processes. In International conference on machine learning (pp. 1704-1713). PMLR.
>
> ***
> Finally, we hope these questions are well answered, and concerns are well addressed. Thanks again for your efforts. For any other question, we are happy to discuss it and make further clarifications.

---

> ### Author Response · Authors · 2023-08-17
> **Update and any other questions**
>
> Dear Reviewer YmWX,
>
> Thanks for your comments. Would you please provide precious feedback on our rebuttal? We are happy to answer any questions.  ***If all concerns are resolved, it would be appreciated to update the score.*** Your update means a lot to us. Thanks.

---

> > ### Comment · Reviewer_YmWX · 2023-08-21
> > **post rebuttal comments**
> >
> > Thanks for authors' efforts for the rebuttal. The response addressed my questions and I'd like to improve 1 point. Thank you.

---

> > > ### Author Response · Authors · 2023-08-21
> > >
> > > Thanks again for your positive comments and kindness.

---

### Author Rebuttal · Authors · 2023-08-08

We sincerely thank all reviewers and area chairs for wonderful work. In this response, I will summarize reviews, address confusion/concerns, and report future changes in the manuscript.
&nbsp;
***
### **I. Positive Comments**

These include: (1) *a novel meta learning pipeline/objective* **[\# Reviewers YmWX/ANpj/vUMr]** to solve interesting problems **[\# Reviewers NA6N]**; (2) *a well-written paper* with good presentation **[\# Reviewers YmWX/ANpj/NA6N/vUMr/Wn6T]**; (3) *extensive experimental results* with good ablations **[\# Reviewers YmWX/ANpj/vUMr]**; (4) lemmas/theorem with well supported proofs **[\# Reviewer ANpj]**.
&nbsp;
***
### **II. Primary Confusion and Concerns**

**1. Require point navigation tasks and DR-CNP results**

Thanks for these constructive comments.

(1) We’ve performed experiments in point robots for navigation tasks [1].
New results as **Table 7**. Meta-testing step-wise rewards in 4 runs in point robot navigation tasks (the higher, the better).

|  Methods     | Average | Worst | $\text{CVaR}_{\alpha}$ |
|--------------|---------|-------|------------------------|
| MAML          | $-16.6(\pm0.19)$     | $-46.9(\pm4.22)$   | $-24.2(\pm0.49)$                    |
| TR-MAML       | $-36.8(\pm0.12)$    | $-81.3(\pm4.35)$   | $-52.1(\pm0.89)$                    |
| DR-MAML(Ours) | $-15.3(\pm0.49)$     | $-46.7(\pm3.24)$   | $-21.6(\pm0.79)$                    |

Please refer to **Response to \# Reviewers Wn6T** and attached PDF for more details and analysis.

(2) We’ve performed experiments in Gaussian Process regression [2] using developed DR-CNPs.
New results as **Table 6**. Meta-testing point-wise log-likelihood results in 5 runs (the higher, the better).

|  Methods     | Average            | Worst              | $\text{CVaR}_{\alpha}$ |
|--------------|--------------------|--------------------|------------------------|
| CNP          | $-0.22(\pm0.04)$      | $-1.35(\pm0.38)$      | $-0.50(\pm0.11)$          |
| TR-CNP       | $-0.80(\pm0.01)$      | $-1.08(\pm0.02)$      | $-0.87(\pm0.01)$          |
| DR-CNP(Ours) | $-0.21(\pm0.02)$ | $-0.74(\pm0.08)$ | $-0.37(\pm0.03)$     |

Please refer to **Response to \# Reviewers YmWX** for implementation and discussions. *Visualized results are in attached PDF files*.

**2. Lack of the Group DRO baseline [3] and discussions [\# Reviewer NA6N]**

Thanks for this advice. ***The Group DRO was in the original submission, corresponding to Appendix K and the reference (Sagawa et al., 2020) [3].***
Please check the contents in *Line 843*:
> Additionally, we use DR-MAML as the example and perform the comparison between our two-stage algorithm and the risk reweighted algorithm (Sagawa et al., 2020)...

**(1) Experimental Results.** The group DRO (Sagawa et al., 2020) [3] experimental results were reported in Table4/5 together with connections. The implementation of the group DRO is the same as (Sagawa et al., 2020) [3]. We’ll rename the DR-MAML (Risk-Reweighted) with DR-MAML (Group DRO).

**(2) Research Motivations.** Ours is similar to the group DRO method, and the difference is that ours focuses on the tail risk only while the group DRO considers the worst group in a risk reweighted way.

*As missing the Group DRO is the major concern, we hope the misunderstanding can be well resolved this time.*

**3. Incremental to TR-MAML and Misconception of Tail Risk [\# Reviewer Wn6T]**

These are contribution clarification and tail risk misconception issues.

**(1) Concerning contributions, our work differs from TR-MAML a lot**: (a) define distributionally robustness for fast adaptation rather than worst-case in TR-MAML; (b) heuristic two-stage optimization algorithms with improvement guarantee; (c) increased robustness without sacrificing the average performance in contrast to TR-MAML. Hence, *the method is new with theoretical contributions and can improve adaptation robustness*. Though the experimental set-up is incremental w.r.t. TR-MAML, we provided extensive new experimental results, e.g., DR-CNP and Meta RL, after taking advice.

**(2) Concerning the tail risk concept in this paper**, it is from the literature work [4-5] and the “tail risk” Wikipedia. We hope such a misconception is addressed.

Other answers to questions, clarifications, and misconceptions are attached in separate reviewers’ responses.

**Reference**

[1] Finn, C., Abbeel, P, & Levine, S. (2017). Model-agnostic meta-learning for fast adaptation of deep networks. ICML.

[2] Garnelo, M., Rosenbaum, D, Maddison, C., Ramalho, T, Saxton, D., Shanahan, M., ... & Eslami, S. A. (2018). Conditional neural processes. ICML.

[3] Sagawa, S., Koh, P. W., Hashimoto, T. B., and Liang, P. (2020). Distributionally robust neural networks. ICLR.

[4] Sarykalin, S., Serraino, G., & Uryasev, S. (2008). Value-at-risk vs. conditional value-at-risk in risk management and optimization. Informs Tutorials in Operations Research.

[5] Hong, L J, & Liu, G. (2009). Simulating sensitivities of conditional value at risk. Management Science.
&nbsp;
***
### **III. Future Changes in the Manuscript**

**1. Additional experimental results**

In Appendix K, *we’ll include the above results and discussions*.

**2. Rename DR-MAML (Risk-Reweighted) with DR-MAML (Group DRO) in Table4/5**

We’ll rename the baseline and move this result to the main paper.

**2. More discussions in literature work [\# Reviewer vUMr]**

We’ll cite and discuss references, see Response to \# Reviewer vUMr for details.

**3. Move the appendix contents to the main paper**

We’ll move Pseudocode in Appendix B/Table4/Table5 result analysis to the main paper.

**4. Typo revisions**

We revised “plaint” to “plain” in Line34; “sometime” to sometimes” in Line 199; a to an in Line 208, Figure4 caption “DR-MAMAL” to “DR-MAML” in Figure4 caption; and Appendix “Risk-Rweighted” to “Risk Reweighted” in Line 859.
&nbsp;
***
Once again, thank all reviewers and area chairs and your effort means a lot in improving our manuscript.

---

### Decision · Program_Chairs · 2023-09-21

**Decision:**

Accept (poster)

**Comment:**

The paper addresses tail risk in meta-learning, i.e. the performance on difficult tasks, by optimizing an approximation of conditional value at risk. The practical algorithm contains two stages, one for estimating the threshold for conditional value at risk via meta-batches, followed by gradient descent. The practical experiments show the efficacy of the proposed method.

The reviewers initially raised several issues regarding the methods’ motivation and requested additional experiments on additional methods/settings. After rebuttal, most concerns are addressed and all reviewers are in favor of acceptance.